# FEDERATED LEARNING AS VARIATIONAL INFERENCE: A SCALABLE EXPECTATION PROPAGATION APPROACH

**Han Guo**[†★] **Philip Greengard**[‡] **Hongyi Wang**[†] **Andrew Gelman**[‡] **Yoon Kim**[★] **Eric P. Xing**[†◇]

[†]Carnegie Mellon University, [★]Massachusetts Institute of Technology, [‡]Columbia University
[◇]Mohamed bin Zayed University of Artificial Intelligence, Petuum Inc.

## ABSTRACT

The canonical formulation of federated learning treats it as a distributed optimization problem where the model parameters are optimized against a global loss function that decomposes across client loss functions. A recent alternative formulation instead treats federated learning as a distributed inference problem, where the goal is to infer a global posterior from partitioned client data (Al-Shedivat et al., 2021). This paper extends the inference view and describes a variational inference formulation of federated learning where the goal is to find a global variational posterior that well-approximates the true posterior. This naturally motivates an expectation propagation approach to federated learning (FedEP), where approximations to the global posterior are iteratively refined through probabilistic message-passing between the central server and the clients. We conduct an extensive empirical study across various algorithmic considerations and describe practical strategies for scaling up expectation propagation to the modern federated setting. We apply FedEP on standard federated learning benchmarks and find that it outperforms strong baselines in terms of both convergence speed and accuracy.[1]

## 1 INTRODUCTION AND BACKGROUND

Many applications of machine learning require training a centralized model over decentralized, heterogeneous, and potentially private datasets. For example, hospitals may be interested in collaboratively training a model for predictive healthcare, but privacy rules might require each hospital's data to remain local. Federated Learning (FL, McMahan et al., 2017; Kairouz et al., 2021; Wang et al., 2021) has emerged as a privacy-preserving training paradigm that does not require clients' private data to leave their local devices. FL introduces new challenges on top of classic distributed learning: expensive communication, statistical/hardware heterogeneity, and data privacy (Li et al., 2020a).

The canonical formulation of FL treats it as a distributed optimization problem where the model parameters $\theta$ are trained on $K$ (potentially private) datasets $\mathcal{D} = \bigcup_{k \in [K]} \mathcal{D}_k$,

$$\theta = \arg\min_{\theta} L(\theta), \quad \text{where} \quad L(\theta) = \sum_{k \in [K]} -\log p(\mathcal{D}_k \mid \theta).$$

Standard distributed optimization algorithms (e.g., data-parallel SGD) are too communication-intensive to be practical under the FL setup. Federated Averaging (FedAvg, McMahan et al., 2017) reduces communication costs by allowing clients to perform multiple local SGD steps/epochs before the parameter updates are sent back to the central server and aggregated. However, due to client data heterogeneity, more local computations could lead to stale or biased client updates, and hence sub-optimal behavior (Charles & Konečnỳ, 2020; Woodworth et al., 2020; Wang et al., 2020a).

An alternative approach is to consider a *Bayesian* formulation of the FL problem (Al-Shedivat et al., 2021). Here, we are interested in estimating the posterior of parameters $p(\theta \mid \mathcal{D})$ given a prior $p(\theta)$ (such as an improper uniform or a Gaussian prior) and a collection of client likelihoods $p(\mathcal{D}_k \mid \theta)$ that are independent given the model parameters,

$$p(\theta \mid \mathcal{D}) \propto p(\theta) \prod_{k \in [K]} p(\mathcal{D}_k \mid \theta).$$

In this case the posterior naturally factorizes across partitioned client data, wherein the *global* posterior equates to a multiplicative aggregate of *local* factors (and the prior). However, exact posterior inference is in general intractable for even modestly-sized models and datasets and requires approx-

---

[1]Code: `https://github.com/HanGuo97/expectation-propagation`. This work was completed while Han Guo was a visiting student at MIT.

imate inference techniques. In this paper we turn to variational inference, in effect transforming the federated optimization problem into a distributed inference problem. Concretely, we view the solution of federated learning as the mode of a variational (posterior) distribution $q \in \mathcal{Q}$ with some divergence function $D(\cdot \| \cdot)$ (e.g., KL-divergence),

$$\boldsymbol{\theta} = \arg\max_{\boldsymbol{\theta}} q(\boldsymbol{\theta}), \quad \text{where } q(\boldsymbol{\theta}) = \arg\min_{q \in \mathcal{Q}} D\left(p\left(\boldsymbol{\theta} \mid \mathcal{D}\right) \| q\left(\boldsymbol{\theta}\right)\right). \tag{1}$$

Under this approach, clients use local computation to perform posterior inference (instead of parameter/gradient estimation) in parallel. In exchange, possibly fewer lockstep synchronization and communication steps are required between clients and servers.

One way to operationalize Eq. 1 is through federated posterior averaging (FedPA, Al-Shedivat et al., 2021), where each client independently runs an approximate inference procedure and then sends the local posterior parameters to the server to be multiplicatively aggregated. However, there is no guarantee that independent approximations to local posteriors will lead to a good global approximate posterior. Motivated by the rich line of work on variational inference on streaming/partitioned data (Broderick et al., 2013; Vehtari et al., 2020), this work instead considers an *expectation propagation* (EP, Minka, 2001) approach to FL. In EP, each partition of the data maintains its own local contribution to the global posterior that is iteratively refined through probabilistic message-passing. When applied to FL, this results in an intuitive training scheme where at each round, each client (1) receives the current approximation to the global posterior from the centralized server, (2) carries out local inference to update its local approximation, and (3) sends the refined approximation to the server to be aggregated. Conceptually, this federated learning with expectation propagation (FedEP) approach extends FedPA by taking into account the current global approximation in step (2).

However, scaling up classic expectation propagation to the modern federated setting is challenging due to the high dimensionality of model parameters and the large number of clients. Indeed, while there is some existing work on expectation propagation-based federated learning (Corinzia et al., 2019; Kassab & Simeone, 2022; Ashman et al., 2022), they typically focus on small models (fewer than 100K parameters) and few clients (at most 100 clients). In this paper we conduct an extensive empirical study across various algorithmic considerations to scale up expectation propagation to contemporary benchmarks (e.g., models with many millions of parameters and datasets with hundreds of thousands of clients). When applied on top of modern FL benchmarks, our approach outperforms strong FedAvg and FedPA baselines.

## 2 FEDERATED LEARNING WITH EXPECTATION PROPAGATION

The probabilistic view from Eq. 1 motivates an alternative formulation of federated learning based on variational inference. First observe that the global posterior $p\left(\boldsymbol{\theta} \mid \mathcal{D}\right)$ given a collection of datasets $\mathcal{D} = \bigcup_{k \in [K]} \mathcal{D}_k$ factorizes as,

$$p\left(\boldsymbol{\theta} \mid \mathcal{D}\right) \propto p(\boldsymbol{\theta}) \prod_{k=1}^{K} p(\mathcal{D}_k \mid \boldsymbol{\theta}) = \prod_{k=0}^{K} p_k(\boldsymbol{\theta}),$$

where for convenience we define $p_0(\boldsymbol{\theta}) := p(\boldsymbol{\theta})$ to be the prior and further use $p_k(\boldsymbol{\theta}) := p(\mathcal{D}_k \mid \boldsymbol{\theta})$ to refer to the local likelihood associated with $k$-th data partition. To simplify notation we hereon refer to the global posterior as $p_{\text{global}}(\boldsymbol{\theta})$ and drop the conditioning on $\mathcal{D}$. Now consider an approximating global posterior $q_{\text{global}}(\boldsymbol{\theta})$ that admits the same factorization as the above, i.e., $q_{\text{global}}(\boldsymbol{\theta}) \propto \prod_{k=0}^{K} q_k(\boldsymbol{\theta})$. Plugging in these terms into Eq. 1 gives the following objective,

$$\arg\max_{\boldsymbol{\theta}} \prod_{k=0}^{K} q_k(\boldsymbol{\theta}), \text{ where } \{q_k(\boldsymbol{\theta})\}_{k=0}^{K} = \arg\min_{q_k \in \mathcal{Q}} D\left(\prod_{k=0}^{K} p_k(\boldsymbol{\theta}) \| \prod_{k=0}^{K} q_k(\boldsymbol{\theta})\right). \tag{2}$$

Here $\mathcal{Q}$ is the variational family, which is assumed to be the same for all clients. This global objective is in general intractable; evaluating $\prod_k p_k(\boldsymbol{\theta})$ requires accessing all clients' data and violates the standard FL assumption. This section presents a probabilistic message-passing algorithm based on expectation propagation (EP, Minka, 2001).

### 2.1 EXPECTATION PROPAGATION

EP is an iterative algorithm in which an intractable target density $p_{\text{global}}(\boldsymbol{\theta})$ is approximated by a tractable density $q_{\text{global}}(\boldsymbol{\theta})$ using a collection of localized inference procedures. In EP, each local inference problem is a function of just $p_k$ and the current global estimate, making it appropriate for the FL setting.

**Algorithm 1** Federated Learning as Inference

1: **for** round $t = 1, \ldots, T$ **do**
2:     Sample a subset of clients $\mathcal{K}$.
3:     **Broadcast** $q_{\text{global}}(\boldsymbol{\theta})$ to the selected clients.
4:     **for** each client $k \in \mathcal{K}$ **in parallel do**
5:         $\Delta q_k(\boldsymbol{\theta}) \leftarrow \text{ClientInfer}(q_{\text{global}}(\boldsymbol{\theta}))$
6:     **end for**
7:     **Collect** $\Delta q_k(\boldsymbol{\theta})$ from the selected clients.
8:     $q_{\text{global}}(\boldsymbol{\theta}) \leftarrow \text{ServerInfer}(\{\Delta q_k(\boldsymbol{\theta})\}_k)$
9: **end for**
10: **Return** $\boldsymbol{\mu}_{\text{global}}$.

---

**Algorithm 2** Approximate Inference: MCMC

1: **Input:** $q_{\backslash k}(\boldsymbol{\theta}; \mathcal{D}_k, \boldsymbol{\eta}_{-k}, \boldsymbol{\Lambda}_{-k})$
2: $\mathcal{S}_k \leftarrow \{\}$
3: **for** $i = 1, \ldots, N$ **do**
4:     $\boldsymbol{\theta}_k^{(i)} \leftarrow \text{SGDEpoch}(-\log q_{\backslash k}, \boldsymbol{\theta}_k^{(i-1)})$
5:     $\mathcal{S}_k \leftarrow \mathcal{S}_k \cup \boldsymbol{\theta}_k^{(i)}$
6: **end for**
7: $\boldsymbol{\eta}_{\backslash k}, \boldsymbol{\Lambda}_{\backslash k} \leftarrow \text{EstimateMoments}(\mathcal{S}_k)$
8: **Output:** $\widehat{q}_{\backslash k}(\boldsymbol{\theta}; \boldsymbol{\eta}_{\backslash k}, \boldsymbol{\Lambda}_{\backslash k})$

---

**Algorithm 3** Gaussian EP: Server Inference

1: **Receive:** $\{\Delta q_k(\boldsymbol{\theta}; \Delta \boldsymbol{\eta}_k, \Delta \boldsymbol{\Lambda}_k)\}_k$
2: $q_{\text{global}}^{\text{new}} \propto q_{\text{global}} \prod_k (\Delta q_k)^\delta$  // Sec. 2.2.3
    $\boldsymbol{\eta}_{\text{global}} \leftarrow \boldsymbol{\eta}_{\text{global}} + \delta \, \text{ServerOptim}(\sum_k \Delta \boldsymbol{\eta}_k)$
    $\boldsymbol{\Lambda}_{\text{global}} \leftarrow \boldsymbol{\Lambda}_{\text{global}} + \delta \, \text{ServerOptim}(\sum_k \Delta \boldsymbol{\Lambda}_k)$
3: **Send:** $q_{\text{global}}(\boldsymbol{\theta}; \boldsymbol{\eta}_{\text{global}}, \boldsymbol{\Lambda}_{\text{global}})$

---

**Algorithm 4** Gaussian EP: Client Inference

1: **Receive:** $q_{\text{global}}(\boldsymbol{\theta}; \boldsymbol{\eta}_{\text{global}}, \boldsymbol{\Lambda}_{\text{global}})$
2: $q_{-k} \propto q_{\text{global}} / q_k$  // cavity distribution
    $\boldsymbol{\eta}_{-k} \leftarrow \boldsymbol{\eta}_{\text{global}} - \boldsymbol{\eta}_k, \qquad \boldsymbol{\Lambda}_{-k} \leftarrow \boldsymbol{\Lambda}_{\text{global}} - \boldsymbol{\Lambda}_k$
3: $\widehat{q}_{\backslash k} \approx q_{\backslash k} \propto p_k \, q_{-k}$  // tilted inference (Sec. 2.2.2)
    $\boldsymbol{\eta}_{\backslash k}, \boldsymbol{\Lambda}_{\backslash k} \leftarrow \text{ApproxInference}(q_{\backslash k} \propto p_k \, q_{-k})$
4: $\Delta q_k \propto \widehat{q}_{\backslash k} / q_{\text{global}}$  // client deltas (Sec. A.1)
    $\Delta \boldsymbol{\eta}_k \leftarrow \boldsymbol{\eta}_{\backslash k} - \boldsymbol{\eta}_{\text{global}}, \qquad \Delta \boldsymbol{\Lambda}_k \leftarrow \boldsymbol{\Lambda}_{\backslash k} - \boldsymbol{\Lambda}_{\text{global}}$
5: $q_k^{\text{new}} \propto q_k (\Delta q_k)^\delta$  // local update (Sec. 2.2.3)
    $\boldsymbol{\eta}_k \leftarrow \boldsymbol{\eta}_k + \delta \, \text{ClientOptim}(\Delta \boldsymbol{\eta}_k)$
    $\boldsymbol{\Lambda}_k \leftarrow \boldsymbol{\Lambda}_k + \delta \, \text{ClientOptim}(\Delta \boldsymbol{\Lambda}_k)$
6: **Send:** $\Delta q_k(\boldsymbol{\theta}; \Delta \boldsymbol{\eta}_k, \Delta \boldsymbol{\Lambda}_k)$

---

Concretely, EP iteratively solves the following problem (either in sequence or parallel),

$$q_k^{\text{new}}(\boldsymbol{\theta}) = \underset{q \in \mathcal{Q}}{\arg\min} \, D\left( \underbrace{p_k(\boldsymbol{\theta}) \, q_{-k}(\boldsymbol{\theta})}_{\propto \, q_{\backslash k}(\boldsymbol{\theta})} \, \| \, \underbrace{q(\boldsymbol{\theta}) \, q_{-k}(\boldsymbol{\theta})}_{\propto \, \widehat{q}_{\backslash k}(\boldsymbol{\theta})} \right), \quad \text{where } q_{-k}(\boldsymbol{\theta}) \propto \frac{q_{\text{global}}(\boldsymbol{\theta})}{q_k(\boldsymbol{\theta})}. \tag{3}$$

Here $q_{\text{global}}(\boldsymbol{\theta})$ and $q_k(\boldsymbol{\theta})$ are the global/local distributions from the current iteration. (See Sec. A.2 for further details). In the EP literature, $q_{-k}(\boldsymbol{\theta})$ is referred to as the *cavity distribution* and $q_{\backslash k}(\boldsymbol{\theta})$ and $\widehat{q}_{\backslash k}(\boldsymbol{\theta})$ are referred to as the target/approximate *tilted distributions*. EP then uses $q_k^{\text{new}}(\boldsymbol{\theta})$ to derive $q_{\text{global}}^{\text{new}}(\boldsymbol{\theta})$. While the theoretical properties of EP are still not well understood (Minka, 2001; Dehaene & Barthelmé, 2015; 2018), it has empirically been shown to produce good posterior approximations in many cases (Li et al., 2015; Vehtari et al., 2020). When applied to FL, the central server initiates the update by sending the parameters of the current global approximation $q_{\text{global}}(\boldsymbol{\theta})$ as messages to the subset of clients $\mathcal{K}$. Upon receiving these messages, each client updates the respective local approximation $q_k^{\text{new}}(\boldsymbol{\theta})$ and sends back the changes in parameters as messages, which is then aggregated by the server. Algorithms 1-4 illustrate the probabilistic message passing with the Gaussian variational family in more detail.

**Remark.** Consider the case where we set $q_{-k}(\boldsymbol{\theta}) \propto 1$ (i.e., an improper uniform distribution that ignores the current estimate of the global parameters). Then Eq. 3 reduces to federated learning with posterior averaging (FedPA) from Al-Shedivat et al. (2021), $q_k^{\text{new}}(\boldsymbol{\theta}) = \arg\min_{q \in \mathcal{Q}} D\left(p_k(\boldsymbol{\theta}) \| q(\boldsymbol{\theta})\right)$. Hence, FedEP improves upon FedPA by taking into account the global parameters and the previous local estimate while deriving the local posterior.[2]

## 2.2 SCALABLE EXPECTATION PROPAGATION

While federated learning with expectation propagation is conceptually straightforward, scaling up FedEP to modern models and datasets is challenging. For one, the high dimensionality of the parameter space of contemporary models can make local inference difficult even with simple mean-field Gaussian variational families. This is compounded by the fact that classic expectation propagation is *stateful* and therefore requires that each client always maintains its local contribution to the global posterior. These factors make classic EP potentially an unideal approach in settings where

---

[2]When the parameters of $q_{\text{global}}(\boldsymbol{\theta})$ and $q_k(\boldsymbol{\theta})$'s are initialized as improper uniform distributions, the first round (but only the first round) of FedEP and FedPA is identical.

the clients may be resource-constrained and/or the number of clients is large enough that each client is updated only a few times during the course of training. This section discusses various algorithmic consideration when scaling up FedEP to contemporary federated learning benchmarks.

### 2.2.1 VARIATIONAL FAMILY

Following prior work on variational inference in high-dimensional parameter space (Graves, 2011; Blundell et al., 2015; Zhang et al., 2019; Osawa et al., 2019), we use the mean-field Gaussian variational family for $\mathcal{Q}$, which corresponds to multivariate Gaussian distributions with diagonal covariance. Although non-diagonal extensions are possible (e.g., through shrinkage estimators (Ledoit & Wolf, 2004)), we empirically found the diagonal to work well while being simple and communication-efficient. For notational simplicity, we use the following two parameterizations of a Gaussian distribution interchangeably,

$$q(\boldsymbol{\theta}) = \mathcal{N}(\boldsymbol{\theta}; \boldsymbol{\mu}, \boldsymbol{\Sigma}) = \mathcal{N}(\boldsymbol{\theta}; \boldsymbol{\eta}, \boldsymbol{\Lambda}), \quad \text{where } \boldsymbol{\Lambda} := \boldsymbol{\Sigma}^{-1}, \boldsymbol{\eta} := \boldsymbol{\Sigma}^{-1}\boldsymbol{\mu}.$$

Conveniently, both products and quotients of Gaussian distributions—operations commonly used in EP—result in another Gaussian distribution, which simplifies the calculation of the cavity distribution $q_{-k}(\boldsymbol{\theta})$ and the global distribution $q_{\text{global}}(\boldsymbol{\theta})$.[3]

### 2.2.2 CLIENT INFERENCE

At each round of training, each client must estimate $\widehat{q}_{\backslash k}(\boldsymbol{\theta})$, its own approximation to the tilted distribution $q_{\backslash k}(\boldsymbol{\theta})$ in Eq. 3. We study various approaches for this estimation procedure.

**Stochastic Gradient Markov Chain Monte Carlo (SG-MCMC).** SG-MCMC (Welling & Teh, 2011; Ma et al., 2015) uses stochastic gradients to approximately sample from local posteriors. We follow Al-Shedivat et al. (2021) and use a simple variant of SGD-based SG-MCMC, where we collect a single sample per epoch to obtain a set of samples $\mathcal{S}_k = \{\boldsymbol{\theta}_k^{(1)}, \dots, \boldsymbol{\theta}_k^{(N)}\}$.[4] The SGD objective in this case is the unnormalized tilted distribution,

$$\underbrace{-\sum_{z \in \mathcal{D}_k} \log p(\boldsymbol{z} \mid \boldsymbol{\theta})}_{-\log p_k(\boldsymbol{\theta})} + \underbrace{\frac{1}{2}\boldsymbol{\theta}^\top \boldsymbol{\Lambda}_{-k}\boldsymbol{\theta} - \boldsymbol{\eta}_{-k}^\top \boldsymbol{\theta}}_{-\log q_{-k}(\boldsymbol{\theta})},$$

which is simply the client negative log likelihood ($-\log p_k(\boldsymbol{\theta})$) plus a regularizer that penalizes parameters that have low probability under the cavity distribution ($-\log q_{-k}(\boldsymbol{\theta})$). This connection makes it clear that the additional client computation compared to FedAvg (which just minimizes the client negative log-likelihood) is negligible. Given a set of samples $\mathcal{S}_k$ from SG-MCMC, we estimate the parameters of the tilted distribution $q_{\backslash k}(\boldsymbol{\theta})$ with moment matching, i.e.,

$$q_{\backslash k}(\boldsymbol{\theta}) = \mathcal{N}(\boldsymbol{\theta}; \boldsymbol{\mu}_{\backslash k}, \boldsymbol{\Sigma}_{\backslash k}) \quad \text{where } \boldsymbol{\mu}_{\backslash k}, \boldsymbol{\Sigma}_{\backslash k} \leftarrow \text{MomentEstimator}(\mathcal{S}_k).$$

While the mean obtained from $\mathcal{S}_k$ via averaging empirically worked well, the covariance estimation was sometimes unstable. We next discuss three alternative techniques for estimating the covariance.

**SG-MCMC with Scaled Identity Covariance.** Our simplest approach approximates the covariance as a scaled identity matrix with a tunable hyper-parameter $\alpha_{\text{cov}}$, i.e., $\boldsymbol{\Sigma}_{\backslash k} \leftarrow \alpha_{\text{cov}}\boldsymbol{I}$. This cuts down the communication cost in half since we no longer have to send messages for the covariance parameters. While extremely simple, we found scaled identity covariance to work well in practice.

**Laplace Approximation.** Laplace's method approximates the covariance as the inverse Hessian of the negative log-likelihood at the (possibly approximate) MAP estimate. Since the exact inverse Hessian is intractable, we follow common practice and approximate it with the diagonal Fisher,

$$\boldsymbol{\Sigma}_{\backslash k} \leftarrow \left(\boldsymbol{H}_k + \boldsymbol{\Sigma}_{-k}^{-1}\right)^{-1}, \text{ where } \boldsymbol{H}_k \approx \text{diag}\underbrace{\left(\mathbb{E}_{x \sim \mathcal{D}_k, y \sim p(y|x,\boldsymbol{\theta})}\left[\left(\nabla_\theta \log p(y \mid \boldsymbol{\theta}, x)\right)^2\right]\right)}_{\text{diagonal Fisher approximation}}.$$

$$(4)$$

---

[3]Specifically, we have the following identities,

$$\mathcal{N}(\boldsymbol{\theta}; \boldsymbol{\eta}_1, \boldsymbol{\Lambda}_1) \mathcal{N}(\boldsymbol{\theta}; \boldsymbol{\eta}_2, \boldsymbol{\Lambda}_2) \propto \mathcal{N}(\boldsymbol{\theta}; \boldsymbol{\eta}_1 + \boldsymbol{\eta}_2, \boldsymbol{\Lambda}_1 + \boldsymbol{\Lambda}_2), \quad \frac{\mathcal{N}(\boldsymbol{\theta}; \boldsymbol{\eta}_1, \boldsymbol{\Lambda}_1)}{\mathcal{N}(\boldsymbol{\theta}; \boldsymbol{\eta}_2, \boldsymbol{\Lambda}_2)} \propto \mathcal{N}(\boldsymbol{\theta}; \boldsymbol{\eta}_1 - \boldsymbol{\eta}_2, \boldsymbol{\Lambda}_1 - \boldsymbol{\Lambda}_2).$$

[4]Unlike Al-Shedivat et al. (2021), we do not apply Polyak averaging (Mandt et al., 2017; Maddox et al., 2019) as we did not find it to improve results in our case.

This approach samples the input $x$ from the data $D_k$ and the output $y$ from the current model $p(y \mid x, \boldsymbol{\theta})$, as recommended by Kunstner et al. (2019). The Fisher approximation requires additional epochs of backpropagation on top of usual SGD (usually 5 in our case), which requires additional client compute.

**Natural Gradient Variational Inference.** Our final approach uses natural-gradient variational inference (NGVI, Zhang et al., 2018; Khan et al., 2018; Osawa et al., 2019), which incorporates the geometry of the distribution to enable faster convergence. Most existing work on NGVI assume a zero-mean isotropic Gaussian prior. We extend NGVI to work with arbitrary Gaussian priors—necessary for regularizing towards the cavity distribution in FedEP. Specifically, NGVI iteratively computes the following for $t = 1 \ldots T_{\mathrm{NGVI}}$ and learning rate $\beta_{\mathrm{NGVI}}$,

$$\boldsymbol{\Sigma}_{\backslash k, t} \leftarrow \left( |\mathcal{D}_k| \boldsymbol{s}_t + \boldsymbol{\Sigma}_{-k}^{-1} \right)^{-1}, \text{where } \boldsymbol{s}_t \leftarrow \beta_{\mathrm{NGVI}} \boldsymbol{s}_{t-1} + (1 - \beta_{\mathrm{NGVI}}) \, \mathbb{E}_{\boldsymbol{\theta} \sim q_{\backslash k, t-1}} \left[ \frac{1}{|\mathcal{D}_k|} \mathrm{Fisher}(\boldsymbol{\theta}) \right].$$

Here $\mathrm{Fisher}(\cdot)$ is the diagonal Fisher approximation in Eq. 4 but evaluated at a sample of parameters from $q_{\backslash k, t}(\boldsymbol{\theta})$, the approximate posterior using the current estimate of $\boldsymbol{\Sigma}_{\backslash k, t}$. We give the exact NGVI update (which is algorithmically similar to the Adam optimizer (Kingma & Ba, 2015)) in Algorithm 5 in the appendix.

### 2.2.3 ADAPTIVE OPTIMIZATION AS DAMPING

Given the approximate tilted distribution $\widehat{q}_{\backslash k}(\boldsymbol{\theta})$ and the corresponding parameters $\boldsymbol{\mu}_{\backslash k}, \boldsymbol{\Sigma}_{\backslash k}$, we can in principle follow the update equation in Eq. 2 to estimate $q_{\mathrm{global}}^{\mathrm{new}}(\boldsymbol{\theta})$. However, adaptive optimizers have been shown to be crucial for scaling federated learning to practical settings (Reddi et al., 2020), and the vanilla EP update does not immediately lend itself to adaptive updates. This section describes an adaptive extension to EP based on damping, in which we to re-interpret a damped EP update as a gradient update on the natural parameters, which allows for the use of adaptive optimizers.

Damping performs client updates only partially with step size $\delta$ and is commonly used in parallel EP settings (Minka & Lafferty, 2002; Vehtari et al., 2020). Letting $\Delta q_k(\boldsymbol{\theta}) \propto \widehat{q}_{\backslash k}(\boldsymbol{\theta}) / q_{\mathrm{global}}(\boldsymbol{\theta})$ denote the client "update" distribution, we can simplify the update and arrive at the following intuitive form (Vehtari et al., 2020) (see Sec. A.1 for derivation),

**Client:** $\quad q_k^{\mathrm{new}}(\boldsymbol{\theta}) \propto q_k(\boldsymbol{\theta}) \left( \Delta q_k(\boldsymbol{\theta}) \right)^{\delta}, \quad$ **Server:** $\quad q_{\mathrm{global}}^{\mathrm{new}}(\boldsymbol{\theta}) \propto q_{\mathrm{global}}(\boldsymbol{\theta}) \prod_k \left( \Delta q_k(\boldsymbol{\theta}) \right)^{\delta}.$

Recalling that products of Gaussian distributions yields another Gaussian distribution that simply sums the natural parameters, the damped update for $\boldsymbol{\eta}$ is given by,

**Client:** $\quad \boldsymbol{\eta}_k \leftarrow \boldsymbol{\eta}_k + \delta \Delta \boldsymbol{\eta}_k, \quad$ **Server:** $\quad \boldsymbol{\eta}_{\mathrm{global}} \leftarrow \boldsymbol{\eta}_{\mathrm{global}} + \delta \sum_{k \in \mathcal{K}} \Delta \boldsymbol{\eta}_k.$

(The update on the precision $\boldsymbol{\Lambda}$ is analogous.) By re-interpreting the update distribution $\Delta q_k(\boldsymbol{\theta}; \Delta \boldsymbol{\eta}_k, \Delta \boldsymbol{\Lambda}_k)$ as a "gradient", we can apply off-the-shelf adaptive optimizers ,

**Client:** $\quad \boldsymbol{\eta}_k \leftarrow \boldsymbol{\eta}_k + \delta \, \mathrm{optim}(\Delta \boldsymbol{\eta}_k), \quad$ **Server:** $\quad \boldsymbol{\eta}_{\mathrm{global}} \leftarrow \boldsymbol{\eta}_{\mathrm{global}} + \delta \, \mathrm{optim}\big( \sum_{k \in \mathcal{K}} \Delta \boldsymbol{\eta}_k \big).$

All our FedEP experiments (and the FedAvg and FedPA baselines) employ adaptive optimization.

### 2.2.4 STOCHASTIC EXPECTATION PROPAGATION FOR STATELESS CLIENTS

Clients are typically assumed to be stateful in the classic formulations of expectation propagation. However, there are scenarios in which stateful clients are infeasible (e.g., memory constraints) or even undesirable (e.g., large number of clients who only participate in a few update rounds, leading to stale messages). We thus additionally experiment with a stateless version of FedEP via stochastic expectation propagation (SEP, Li et al., 2015). SEP employs direct iterative refinement of a global approximation comprising the prior $p(\boldsymbol{\theta})$ and $K$ copies of a *single* approximating factor $\overline{q_k}(\boldsymbol{\theta})$,

$$q_{\mathrm{global}}(\boldsymbol{\theta}) \propto p(\boldsymbol{\theta}) \left( \overline{q_k}(\boldsymbol{\theta}) \right)^K.$$

That is, clients are assumed to capture the average effect. In practice, FedSEP is implemented in Algorithm 4 via replacing the cavity update (step 2) with $q_{-k}(\boldsymbol{\theta}) \propto q_{\mathrm{global}}(\boldsymbol{\theta}) / \overline{q_k}(\boldsymbol{\theta})$ and removing the local update (step 5).

| Dataset | Model | Model size | Clients (train/test) | Examples per client (train/test) |
|---------|-------|-----------|---------------------|---------------------------------|
| CIFAR-100 | ResNet-18 | 11.2M | 500 / 100 | $100_{\pm 0}$ / $100_{\pm 0}$ |
| StackOverflow | Linear | 5.0M | 342,477 / 204,088 | $397_{\pm 1279}$ / $81_{\pm 301}$ |
| EMNIST-62 | CNN | 1.2M | 3,400 / 3,400 | $198_{\pm 77}$ / $23_{\pm 9}$ |

**Table 1:** Model and dataset statistics. The $\pm$ in "Examples per client" client denotes standard deviation.

## 3 EXPERIMENTS

We empirically study FedEP across various benchmarks. We start with a toy setting in Sec. 3.1 where we examine cases where federated posterior average (FedPA, Al-Shedivat et al., 2021), which does not take into account global and other clients' approximations during client inference, performs sub-optimally. We then turn to realistic federated learning benchmarks in Sec. 3.2, where both the size of the model and the number of clients are much larger than had been previously considered in prior EP-based approaches to federated learning (Corinzia et al., 2019; Kassab & Simeone, 2022). Here, we resort to the techniques discussed in Sec. 2.2: approximate inference of the tilted distributions, adaptive optimization, and possibly stateless clients. Finally, we conclude in Sec. 3.3 with an analysis of some of the observations from the benchmark experiments.

### 3.1 TOY EXPERIMENTS

We start with a simple toy setting to illustrate the differences between FedPA and FedEP. Here the task is to infer the global mean from two clients, each of which is parameterized as a two-dimensional Gaussian, $p_k(\boldsymbol{\theta}) = \mathcal{N}(\boldsymbol{\theta}; \boldsymbol{\mu}_k, \boldsymbol{\Sigma}_k)$ for $k \in \{1, 2\}$. Assuming an improper uniform prior, the global distribution is then also a Gaussian with its posterior mode coinciding with the global mean. We perform exact inference via analytically solving $D_{\mathrm{KL}}(q_{\setminus k} \| \widehat{q}_{\setminus k})$, but restrict the variational family to Gaussians with diagonal covariance (i.e., mean-field family). In this case both the FedAvg and FedPA solution can be derived in "one-shot". Fig. 1 illustrates a simple case where posterior averaging performs sub-optimally. On the other hand, expectation propagation iteratively refines the approximations toward the globally optimal estimation.

We study this phenomena more systematically by sampling random client distributions, where the client parameters are sampled from the normal-inverse-Wishart (NIW) distribution,

$$\boldsymbol{\mu}_k \sim \mathcal{N}\left(\boldsymbol{\mu} \mid \boldsymbol{\mu}_0, \frac{1}{\lambda}\boldsymbol{\Sigma}_k\right), \quad \boldsymbol{\Sigma}_k \sim \mathcal{W}^{-1}(\boldsymbol{\Sigma} \mid \boldsymbol{\Psi}, \nu).$$

Here we set the hyper-prior mean $\boldsymbol{\mu}_0 = \mathbf{0}$, degrees of freedom $\nu = 7$, scale $\lambda = 0.2$, and sample a random symmetric positive definite matrix for $\boldsymbol{\Psi}$. Table 4 shows the average Euclidean distances between the estimated and target global mean for FedAvg, FedPA, and FedEP averaged over 200 random samples of client distributions. Experimental results demonstrate that iterative message passing in FedEP consistently improves upon the sub-optimal solution from posterior averaging.

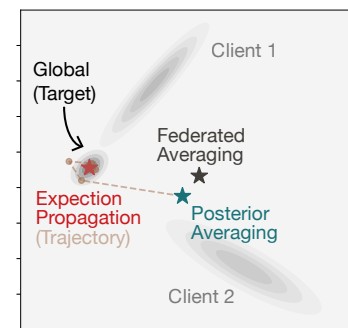

**Figure 1:** FedAvg, FedPA, and FedEP on a toy two dimensional dataset with two clients.

### 3.2 BENCHMARKS EXPERIMENTS

We next conduct experiments on a suite of realistic benchmark tasks introduced by Reddi et al. (2020). Table 1 summarizes the model and raw dataset statistics, which is the same as in Al-Shedivat et al. (2021). We use the dataset preprocessing provided in TensorFlow Federated (TFF, Authors, 2018), and implement the models in Jax (Bradbury et al., 2018; Hennigan et al., 2020; Ro et al., 2021). We compare against both FedAvg with adaptive optimizers and FedPA.[5] As in FedPA, we run a few rounds of FedAvg as burn-in before switching to FedEP. We refer the reader to the appendix for the exact experimental setup.

For evaluation we consider both convergence speed and final performance. On CIFAR-100 and EMNIST-62, we measure the (1) number of rounds to reach certain accuracy thresholds (based on 10-round running averages), and (2) the best accuracy attained within specific rounds (based on 100-round running averages). For StackOverflow, we measure the best precision, recall, micro- and

---

[5]Reddi et al. (2020) refer to federated averaging with adaptive server optimizers as FedAdam etc. We refer to this as FedAvg for simplicity.

| Method | Accuracy (%, ↑) | | Rounds (#, ↓) | |
|---|---|---|---|---|
| | 1000R | 1500R | 45% | 50% |
| FedPA | 45.8 | 48.4 | 811 | – |
| FedAvg | $44.7_{(0.2)}$ | $46.2_{(0.2)}$ | $911_{(86)}$ | – |
| FedEP (I) | $\underline{48.7}_{(0.4)}$ | $\mathbf{50.7}_{(0.4)}$ | $\underline{473}_{(17)}$ | $\mathbf{1167}_{(107)}$ |
| FedEP (M) | $\mathbf{48.8}_{(0.4)}$ | $\underline{50.4}_{(0.5)}$ | $\mathbf{461}_{(13)}$ | $1240^{\dagger}_{(133)}$ |
| FedEP (L) | $46.5_{(0.4)}$ | $47.7_{(0.3)}$ | $523_{(28)}$ | – |
| FedEP (V) | $47.8_{(0.5)}$ | $49.6_{(0.6)}$ | $487_{(24)}$ | $1290^{\ddagger}_{(-)}$ |
| FedSEP (I) | $48.2_{(0.4)}$ | $48.9_{(0.4)}$ | $438_{(9)}$ | – |
| FedSEP (M) | $48.2_{(0.4)}$ | $48.9_{(0.4)}$ | $438_{(9)}$ | – |
| FedSEP (L) | $47.2_{(0.4)}$ | $47.8_{(0.4)}$ | $442_{(10)}$ | – |
| FedSEP (V) | $47.8_{(0.4)}$ | $48.5_{(0.5)}$ | $440_{(10)}$ | – |

**Table 2:** CIFAR-100 Experiments. Statistics shown are the averages and standard deviations (subscript in brackets) aggregated over 5 seeds. We measure the number of rounds to reach certain accuracy thresholds (based on 10-round running averages) and the best accuracy attained within specific rounds (based on 100-round running averages). FedEP and FedSEP refer to the stateful EP and stateless stochastic EP. We use **I** (Scaled Identity Covariance), **M** (MCMC), **L** (Laplace), and **V** (NGVI) to refer to different inference techniques. †One seed does not reach the threshold. ‡Only one seed reaches the threshold.

| Method | prec. | recall | mi-F1 | ma-F1 |
|---|---|---|---|---|
| FedPA | $\underline{74.66}$ | 19.94 | 30.78 | 11.63 |
| FedAvg | $\mathbf{75.20}_{(0.18)}$ | $13.88_{(0.27)}$ | $23.32_{(0.41)}$ | $8.02_{(0.26)}$ |
| FedSEP (I) | $71.32_{(0.20)}$ | $25.10_{(0.22)}$ | $37.04_{(0.25)}$ | $13.61_{(0.15)}$ |
| FedSEP (M) | $58.31_{(18.04)}$ | $8.70_{(1.14)}$ | $14.29_{(2.20)}$ | $2.70_{(0.33)}$ |
| FedSEP (L) | $70.98_{(0.18)}$ | $\underline{25.88}_{(0.30)}$ | $\underline{37.80}_{(0.29)}$ | $\underline{13.97}_{(0.19)}$ |
| FedSEP (V) | $69.51_{(0.35)}$ | $\mathbf{28.02}_{(0.20)}$ | $\mathbf{39.78}_{(0.25)}$ | $\mathbf{15.32}_{(0.08)}$ |

**Table 3:** StackOverflow Experiments. Statistics shown are the averages and standard deviations (subscript in brackets) aggregated over 5 seeds. We measure the best precision (**prec.**), **recall**, micro- and macro-F1 (**mi/ma-F1**) attained by round 1500 (based on 100-round running averages).

| Method | Euclidean Distance |
|---|---|
| FedAvg | $5.4 \times 10^{-1} \pm 4.7 \times 10^{-1}$ |
| FedPA | $2.6 \times 10^{-1} \pm 2.6 \times 10^{-1}$ |
| FedEP | $1.1 \times 10^{-7} \pm 9.8 \times 10^{-8}$ |

**Table 4:** Toy Experiments. Statistics shown are the averages and standard deviations of Euclidean distances between the estimated and target global mean aggregated over 200 random samples of client distributions.

macro-F1 attained by round 1500 (based on 100-round running averages).[6] Due to the size of this dataset, the performance at each round is evaluated on a $10K$ subsample. The evaluation setup is almost exactly the same as in prior work (Reddi et al., 2020; Al-Shedivat et al., 2021). Due to space we mainly discuss the CIFAR-100 ("CIFAR") and StackOverflow Tag Prediction ("StackOverflow") results in this section and defer the EMNIST-62 ("EMNIST") results (which are qualitatively similar) to the appendix (Sec. A.3).

**CIFAR.** In Table 2 and Fig. 2 (left, mid), we compare FedAvg, FedPA, and FedEP with various approaches for approximating the clients' tilted distributions (Sec. 2.2.2). A notable observation is the switch from FedAvg to FedPA/FedEP at the 400th round, where observe significant increases in performance. Somewhat surprisingly, we find that scaled identity is a simple yet strong baseline. (We conduct further experiments in Sec. 3.3 to analyze this phenomena in greater detail). We next experiment with stochastic EP (FedSEP, Sec. 2.2.4), a stateless version of FedEP that is more memory-efficient. We find that FedSEP can almost match the performance of full EP despite being much simpler (Fig. 2, right).

**StackOverflow.** Experiments on CIFAR study the challenges when scaling FedEP to richly parameterized neural models with millions of parameters. Our StackOverflow experiments are on the other hand intended to investigate whether FedEP can scale to regimes with a large number of clients (hundreds of thousands). Under this setup the number of clients is large enough that the average client will likely only ever participate in a single update round, which renders the stateful version of FedEP meaningless. We thus mainly experiment with the stateless version of FedEP.[7] Table 3 and Fig. 3 (full figure available in the appendix Fig. 5) show the results comparing the same set of approximate client inference techniques. These experiments demonstrate the scalability of EP to a large number of clients even when we assume clients are stateless.

### 3.3 ANALYSIS AND DISCUSSION

**The Effectiveness of Scaled Identity.** Why does the scaled identity approximation work so well? We investigate this question in the same toy setting as in Sec. 3.1. Fig. 4 (left) compares the scaled-identity EP with FedEP, FedPA, and FedAvg. Unsurprisingly, this restriction leads to worse performance initially. However, as clients pass messages between each other, scaled-identity EP eventually converges to nearly the same approximation as diagonal EP.

The toy experiments demonstrate the effectiveness of scaled identity in terms of the *final solution*. However, this does not fully explain the benchmark experiments where we observed scaled iden-

---

[6]TFF by default considers a threshold-based precision and top-5 recall. Our early experiments found that threshold-based metrics correlate better with loss, and use them in StackOverflow experiments.

[7]This was also due to the practical difficulty of storing all the clients' distributions.

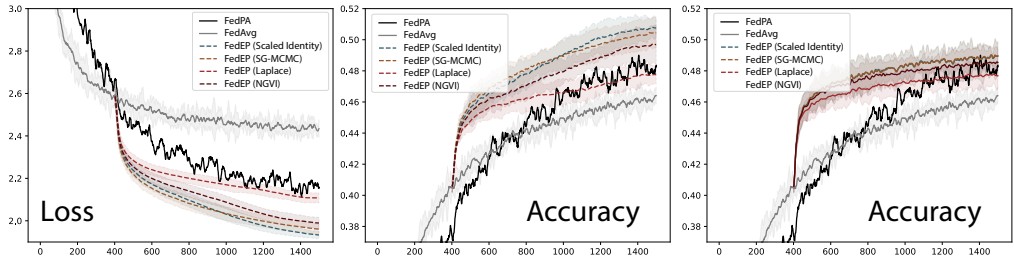

**Figure 2:** CIFAR-100 Experiments. **Left and Middle:** loss and accuracy of the server as a function of rounds for FedAvg, FedPA, and (stateful) FedEP with various inference techniques. **Right:** accuracy as a function of rounds for FedAvg, FedPA, and (stateless) FedSEP. The transitions from FedAvg to FedPA, FedEP, and FedSEP happen at round 400. Lines and shaded regions refer to the averages and 2 standard deviations over 5 runs, resp.

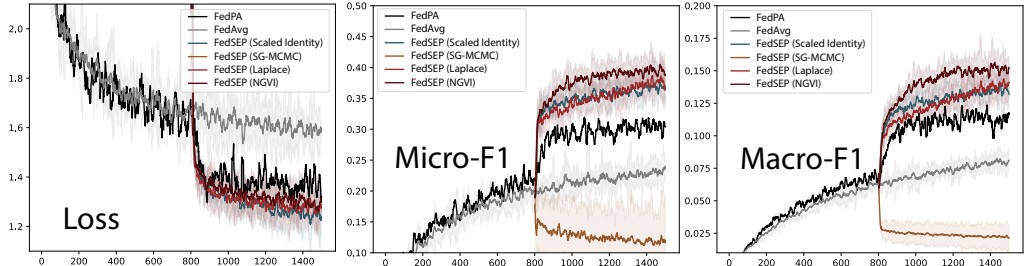

**Figure 3:** StackOverflow Experiments. Curves represent loss, micro-F1, and macro-F1 of the global parameter estimation as a function of rounds for FedAvg, FedPA, and (stateless) FedSEP with various inference techniques. The transitions from FedAvg to FedPA and FedSEP happen at round 800. Lines and shaded regions refer to the averages and 2 standard deviations over 5 runs, resp.

tity EP to match more involved variants in terms of *convergence speed.* We hypothesize that as models grow more complex, the gap between scaled identity and other techniques might decrease due to the difficulty of obtaining credible estimates of (even diagonal) covariance in high dimensional settings. To test this, we revisit the CIFAR-100 task and compare the following two settings: "small" setting which uses a smaller linear model on the PCA'ed features and has $10.1K$ parameters, and a "large" setting that uses a linear model on the raw features and has $172.9K$ parameters. For each setting, we conduct experiments with EP using scaled-identity and NGVI and plot the results in Fig. 4 (right). We observe that under the "small" setting, a more advanced approximate inference technique converges faster than scaled-identity EP, consistent with the toy experiments. As we increase the model size however ("large" setting), the gap between these two approaches disappears. This indicates that as the model gets more complex, the convergence benefits of more advanced approximate inference decline due to covariance estimation's becoming more difficult.

**Uncertainty Quantification.** One motivation for a Bayesian approach is uncertainty quantification. We thus explore whether a Bayesian treatment of federated learning results in models that have better expected calibration error (ECE, Naeini et al., 2015; Guo et al., 2017), which is defined as $\text{ECE} = \sum_i^{N_{\text{bins}}} b_i \left| \text{accuracy}_i - \text{confidence}_i \right|$. Here $\text{accuracy}_i$ is the top-1 prediction accuracy in $i$-th bin, $\text{confidence}_i$ is the average confidence of predictions in $i$-th bin, and $b_i$ is the fraction of data points in $i$-th bin. Bins are constructed in a uniform way in the $[0, 1]$ range.[8] We consider accuracy and calibration from the resulting approximate posterior in two ways: (1) point estimation, which uses the final model (i.e., MAP estimate from the approximate posterior) to obtain the output probabilities for each

| Method | Accuracy (%, ↑) | | ECE-15 (%, ↓) | |
| --- | --- | --- | --- | --- |
| | Point Est. | Marg. | Point Est. | Marg. |
| FedPA | 48.1 | − | 13.6 | − |
| FedAvg | 46.6$_{(0.7)}$ | − | 19.5$_{(0.4)}$ | − |
| FedEP (I) | 50.8$_{(0.4)}$ | 49.6$_{(0.6)}$ | 4.9$_{(0.3)}$ | 7.9$_{(0.2)}$ |
| FedEP (M) | 50.5$_{(0.5)}$ | 50.2$_{(0.4)}$ | 5.9$_{(0.5)}$ | 4.6$_{(0.4)}$ |
| FedEP (L) | 47.7$_{(0.5)}$ | 47.8$_{(0.5)}$ | 8.8$_{(0.4)}$ | 6.6$_{(0.4)}$ |
| FedEP (V) | 49.7$_{(0.5)}$ | 49.5$_{(0.3)}$ | 5.9$_{(0.4)}$ | 2.2$_{(0.5)}$ |
| FedSEP (I) | 49.0$_{(0.4)}$ | 48.5$_{(0.4)}$ | 10.0$_{(0.4)}$ | 3.4$_{(0.3)}$ |
| FedSEP (M) | 48.9$_{(0.4)}$ | 48.6$_{(0.4)}$ | 10.1$_{(0.4)}$ | 3.5$_{(0.3)}$ |
| FedSEP (L) | 47.7$_{(0.5)}$ | 47.8$_{(0.5)}$ | 9.6$_{(0.6)}$ | 7.2$_{(0.6)}$ |
| FedSEP (V) | 48.5$_{(0.4)}$ | 48.7$_{(0.4)}$ | 9.3$_{(0.4)}$ | 3.7$_{(0.4)}$ |

**Table 5:** CIFAR-100 Calibration Experiments. FedEP and FedSEP refer to the stateful EP and stateless stochastic EP. We use **I** (Scaled Identity Covariance), **M** (MCMC), **L** (Laplace), and **V** (NGVI) to refer to different inference techniques.

data point, and (2) marginalized estimation, which samples 10 models from the approximate posterior and averages the output probabilities to obtain the final prediction probability. In Table 5,

---

[8]We also experimented with an alternative binning method which puts an equal number of data points in each bin and observed qualitatively similar results.

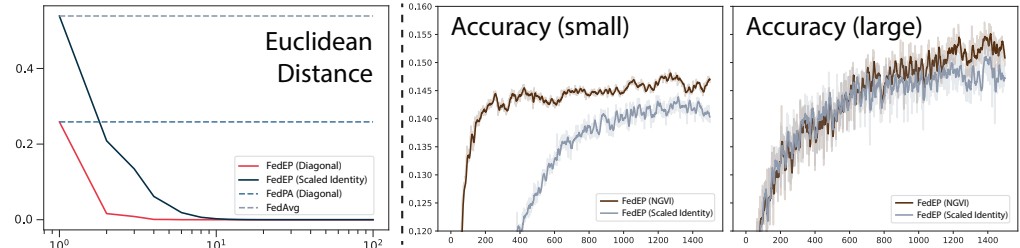

**Figure 4:** Analysis Experiments. **Left:** the average Euclidean distances between the estimated and target global mean as a function of rounds in the toy setting. **Middle and Right:** accuracy as a function of rounds in the CIFAR-100 setting, with either a (relatively) small model (middle) or large model (right).

we observe that FedEP/FedSEP improves both the accuracy (higher is better) as well as expected calibration error (lower is better), with marginalization sometimes helping.

**Hyperparameters.** Table 8 shows the robustness of FedEP w.r.t. various hyperparameters.

**Limitations.** While FedEP outperforms strong baselines in terms of convergence speed and final accuracy, it has several limitations. The stateful variant requires clients to maintain its current contribution to the global posterior, which increases the clients' memory requirements. The non-scaled-identity approaches also impose additional communication overhead due to the need to communicate the diagonal covariance vector. Further, while SG-MCMC/Scaled-Identity approaches have the same compute cost as FedAvg on the client side, Laplace/NGVI approaches require more compute to estimate the Fisher term. Finally, from a theoretical perspective, while the convergence properties of FedAvg under various assumptions have been extensively studied (Li et al., 2018; 2020b), such guarantees for expectation propagation-based approaches remains an open problem.

## 4 RELATED WORK

**Federated Learning.** FL is a paradigm for collaborative learning with decentralized private data (Konečný et al., 2016; McMahan et al., 2017; Li et al., 2020a; Kairouz et al., 2021; Wang et al., 2021). Standard approach to FL tackles it as a distributed optimization problem where the global objective is defined by a weighted combination of clients' local objectives (Mohri et al., 2019; Li et al., 2020a; Reddi et al., 2020; Wang et al., 2020b). Theoretical analysis has demonstrated that federated optimization exhibits convergence guarantees but only under certain conditions, such as a bounded number of local epochs (Li et al., 2020b). Other work has tried to improve the averaging-based aggregations Yurochkin et al. (2019); Wang et al. (2020a). Techniques such as secure aggregation (Bonawitz et al., 2017; 2019; He et al., 2020) and differential privacy (Sun et al., 2019; McMahan et al., 2018) have been widely adopted to further improve privacy in FL (Fredrikson et al., 2015). Our proposed method is compatible with secure aggregation because it conducts server-side reductions over $\Delta\boldsymbol{\eta}_k, \Delta\boldsymbol{\Lambda}_k$.

**Expectation Propagation and Approximate Inference.** This work considers EP as a general technique for passing messages between clients and servers on partitioned data. Here, the cavity distribution "summarizes" the effect of inferences from all other partitions and can be used as a prior in the client's local inference. Historically, EP usually refers to a specific choice of divergence function $D_{\text{KL}}(p\|q)$ (Minka, 2001). This is also known as Variational Message Passing (VMP, Winn et al., 2005) when $D_{\text{KL}}(q\|p)$ is used instead, and Laplace propagation (LP, Smola et al., 2003) when Laplace approximation is used. There have been works that formulate federated learning as a probabilistic inference problem. Most notably, Al-Shedivat et al. (2021) formulate FL as a posterior inference problem. Achituve et al. (2021) apply Gaussian processes with deep kernel learning (Wilson et al., 2016) to personalized FL. Finally, some prior works also consider applying EP to federated learning (Corinzia et al., 2019; Kassab & Simeone, 2022; Ashman et al., 2022), but mostly on relatively small-scale tasks. In this work, we instead discuss and empirically study various algorithmic considerations to scale up expectation propagation to contemporary benchmarks.

## 5 CONCLUSION

This work introduces a probabilistic message-passing algorithm for federated learning based on expectation propagation (FedEP). Messages (probability distributions) are passed to and from clients to iteratively refine global approximations. To scale up classic expectation propagation to the modern FL setting, we discuss and empirically study various algorithmic considerations, such as choice of variational family, approximate inference techniques, adaptive optimization, and stateful/stateless clients. These enable practical EP algorithms for modern-scale federated learning models and data.

**Reproducibility Statement.** For experiment details such as the dataset, model, and hyperparameters, we provide detailed descriptions in Sec. 3 as well as Sec. A.4. We also include in the Appendix additional derivations related to adaptive optimization and damping (Sec. A.1).

## ACKNOWLEDGMENTS

We thank the anonymous reviewers for their comments, and are grateful to Maruan Al-Shedivat for his feedback. EX is supported by NSF IIS1563887, NSF CCF1629559, NSF IIS1617583, NGA HM04762010002, NIGMS R01GM140467, NSF IIS1955532, NSF CNS2008248, NSF IIS2123952, and NSF BCS2040381. YK acknowledges the support of MIT-IBM Watson AI and Amazon.

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

# A APPENDIX

## A.1 DAMPED CLIENT AND SERVER UPDATES

To simplify the notations, observe that Eq. 3 could be re-written in the following way,

$$q_k^{\text{new}}(\boldsymbol{\theta}) \propto \frac{\widehat{q}_{\backslash k}(\boldsymbol{\theta})}{q_{-k}(\boldsymbol{\theta})}, \quad \text{where } \widehat{q}_{\backslash k}(\boldsymbol{\theta}) = \underset{\widehat{q}_{\backslash k} \in \mathcal{Q}}{\arg\min} \, D\Big( p_k(\boldsymbol{\theta}) \, q_{-k}(\boldsymbol{\theta}) \parallel \widehat{q}_{\backslash k}(\boldsymbol{\theta}) \Big).$$

A partially damped client update could be carried out by,

$$
\begin{aligned}
\textbf{Client:} \quad q_k^{\text{new}}(\boldsymbol{\theta}) &\propto \Big( q_k(\boldsymbol{\theta}) \Big)^{1-\delta} \left( \frac{\widehat{q}_{\backslash k}(\boldsymbol{\theta})}{q_{-k}(\boldsymbol{\theta})} \right)^{\delta} \\
&\propto \Big( q_k(\boldsymbol{\theta}) \Big)^{1-\delta} \left( \frac{\widehat{q}_{\backslash k}(\boldsymbol{\theta})}{q_{\text{global}}(\boldsymbol{\theta})/q_k(\boldsymbol{\theta})} \right)^{\delta} \\
&\propto \Big( q_k(\boldsymbol{\theta}) \Big)^{1-\delta} \Big( q_k(\boldsymbol{\theta}) \Big)^{\delta} \left( \frac{\widehat{q}_{\backslash k}(\boldsymbol{\theta})}{q_{\text{global}}(\boldsymbol{\theta})} \right)^{\delta} \\
&\propto q_k(\boldsymbol{\theta}) \left( \frac{\widehat{q}_{\backslash k}(\boldsymbol{\theta})}{q_{\text{global}}(\boldsymbol{\theta})} \right)^{\delta} \\
&\propto q_k(\boldsymbol{\theta}) \Big( \Delta q_k(\boldsymbol{\theta}) \Big)^{\delta},
\end{aligned}
$$

$$\text{where we define} \quad \Delta q_k(\boldsymbol{\theta}) \propto \frac{\widehat{q}_{\backslash k}(\boldsymbol{\theta})}{q_{\text{global}}(\boldsymbol{\theta})}.$$

Similarly, (damped) server updates could be written as the following,

$$
\begin{aligned}
\textbf{Server:} \quad q_{\text{global}}^{\text{new}}(\boldsymbol{\theta}) &\propto \prod_k q_k^{\text{new}}(\boldsymbol{\theta}) \\
&\propto \prod_k q_k(\boldsymbol{\theta}) \Big( \Delta q_k(\boldsymbol{\theta}) \Big)^{\delta} \\
&\propto \left[ \prod_k q_k(\boldsymbol{\theta}) \right] \left[ \prod_k \Big( \Delta q_k(\boldsymbol{\theta}) \Big)^{\delta} \right] \\
&\propto q_{\text{global}}(\boldsymbol{\theta}) \prod_k \Big( \Delta q_k(\boldsymbol{\theta}) \Big)^{\delta}.
\end{aligned}
$$

## A.2 EXPECTATION PROPAGATION (EXTENDED)

Expectation propagation (EP) Minka (2001); Vehtari et al. (2020) constructs a posterior approximation through iterating local computations that refine factors that approximate the posterior contribution from each client. In this spirit, we would ideally like to solve the following localized version of Eq. 2, where we replace one of the factors with its corresponding approximating factor,

$$q_k^{\text{new}}(\boldsymbol{\theta}) = \underset{q \in \mathcal{Q}}{\arg\min} \, D\Big( p_k(\boldsymbol{\theta}) \, p_{-k}(\boldsymbol{\theta}) \parallel q(\boldsymbol{\theta}) \, p_{-k}(\boldsymbol{\theta}) \Big), \quad \text{where } p_{-k}(\boldsymbol{\theta}) \propto \frac{p_{\text{global}}(\boldsymbol{\theta})}{p_k(\boldsymbol{\theta})}.$$

Unfortunately, the right-hand side of the divergence is the intractable posterior we would like to approximate in the first place. Instead, EP solves the following problem (Eq. 3),

$$q_k^{\text{new}}(\boldsymbol{\theta}) = \underset{q \in \mathcal{Q}}{\arg\min} \, D\Big( p_k(\boldsymbol{\theta}) \, q_{-k}(\boldsymbol{\theta}) \parallel q(\boldsymbol{\theta}) \, q_{-k}(\boldsymbol{\theta}) \Big), \quad \text{where } q_{-k}(\boldsymbol{\theta}) \propto \frac{q_{\text{global}}(\boldsymbol{\theta})}{q_k(\boldsymbol{\theta})}.$$

## A.3 ADDITIONAL EXPERIMENTS AND DETAILS

**StackOverflow.** Please see Fig. 5 for additional visualizations.

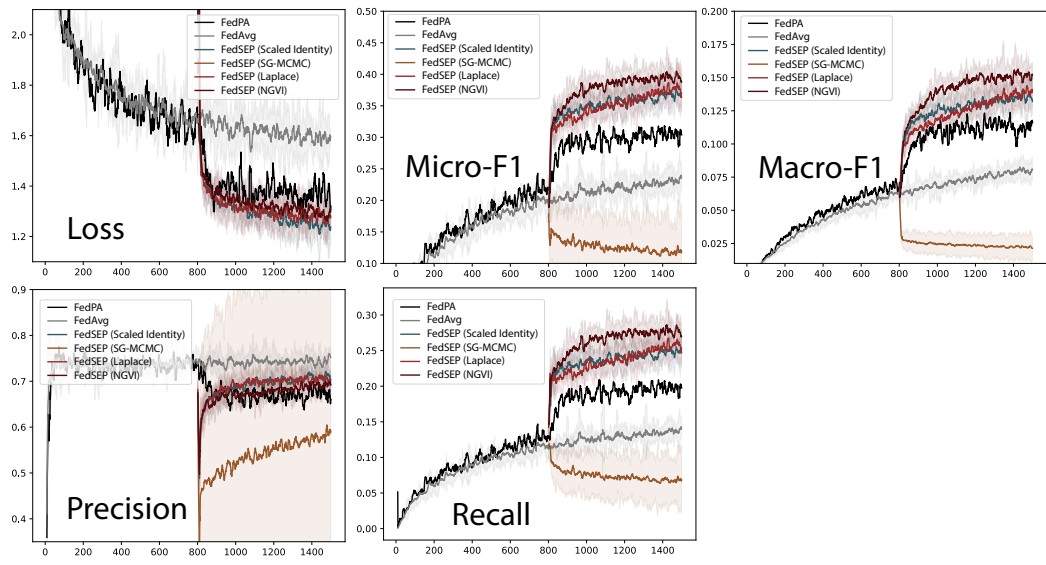

**Figure 5:** Extended StackOverflow Visualizations.

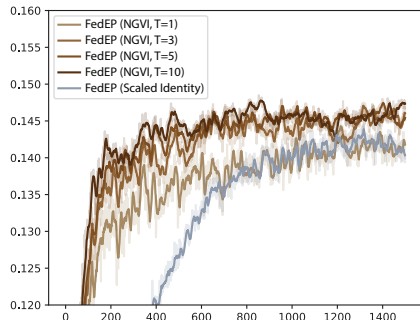

**Figure 6:** Accuracy as a function of rounds, and number of NGVI epochs ($T_{\text{NGVI}}$) in the CIFAR-100 setting, with a (relatively) small model.

**EMNIST.**    Please see Fig. 7 and Table 6 for experimental results.

**Analysis.**    This section extends the experiments (the "small" setting) in Sec. 3.3. It looks at the performance as we increase the complexity (a proxy of quality) of approximate inference techniques. We vary the number of iterations in NGVI from 1 (cheap) to 10 (expensive) epochs. We can observe in Fig. 6 that as we increase NGVI's computations, the performance improves.

## A.4    HYPERPARAMETERS

Please see Table 7 for hyperparameter details. In Table 8, we also conduct experiments to understand their influence on the different algorithms.

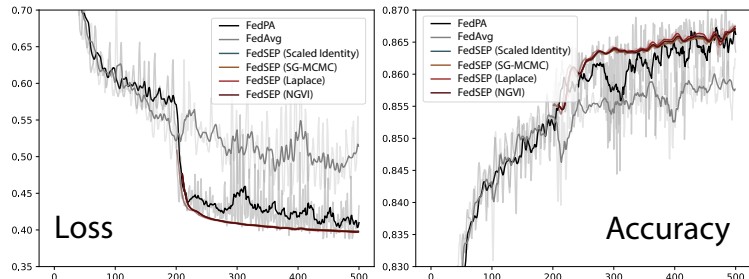

**Figure 7:** EMNIST-62 Experiments. Figures show the loss and accuracy of the global parameter estimation as a function of rounds for FedAvg, FedPA, and (stateless) FedSEP with various inference techniques. The transitions from FedAvg to FedPA and FedSEP happen at round 200.

| Method | accuracy (%, ↑) 300R | accuracy (%, ↑) 500R | rounds (#, ↓) 86% | rounds (#, ↓) 86.5% |
|---|---|---|---|---|
| FedPA | 85.9 | 86.5 | 246 | 398 |
| FedAvg | 85.3 | 85.8 | 465 | – |
| FedSEP (I) | **86.1** | **86.6** | **228** | 399 |
| FedSEP (M) | **86.1** | **86.6** | **228** | 399 |
| FedSEP (L) | **86.1** | **86.6** | **228** | **364** |
| FedSEP (V) | **86.1** | **86.6** | **228** | 382 |

**Table 6:** EMNIST-62 Experiments. We measure the number of rounds to reach certain accuracy thresholds (based on 10-round running averages) and the best accuracy attained within specific rounds (based on 100-round running averages). We use **I** (Scaled Identity Covariance), **M** (MCMC), **L** (Laplace), and **V** (NGVI) to refer to different inference techniques.

---

**Algorithm 5** Approximate Inference: NGVI

---

1: **Input:** $\mathcal{D}_k, \boldsymbol{\mu}_{\backslash k}, \boldsymbol{\Sigma}_{-k}, T_{\text{NGVI}}, N_{\text{NGVI}}, \beta_{\text{NGVI}}$

2: Initialize $\boldsymbol{s}_0, \boldsymbol{\Sigma}_{\backslash k,0}$

3: **for** $t = 1, \ldots, T_{\text{NGVI}}$ **do**

4:     $\mathcal{F} \leftarrow \{\}$

5:     **for** $i = 1, \ldots, N_{\text{NGVI}}$ **do**

6:         $\boldsymbol{\theta} \sim \mathcal{N}\left(\boldsymbol{\theta}; \boldsymbol{\mu}_{\backslash k}, \boldsymbol{\Sigma}_{\backslash k,t-1}\right)$

7:         $\mathcal{F} \leftarrow \mathcal{F} \cup \frac{1}{|\mathcal{D}_k|} \text{Fisher}(\boldsymbol{\theta}, \mathcal{D}_k).$

8:     **end for**

9:     $\boldsymbol{F} \leftarrow \text{Average}(\mathcal{F})$

10:     $\boldsymbol{s}_t \leftarrow \beta_{\text{NGVI}} \boldsymbol{s}_{t-1} + (1 - \beta) \boldsymbol{F}$

11:     $\boldsymbol{\Sigma}_{\backslash k,t} \leftarrow \left(|\mathcal{D}_k|\boldsymbol{s}_t + \boldsymbol{\Sigma}_{-k}^{-1}\right)^{-1}$

12: **end for**

13: **Output:** $\boldsymbol{\Sigma}_{\backslash k, T_{\text{NGVI}}}$

---

| Hyperparameter | CIFAR-100 | StackOverflow | EMNIST-62 |
|---|---|---|---|
| | Task Hyperparameters from Al-Shedivat et al. (2021) | | |
| Server Optimizer | SGD ($m = 0.9$) | Adagrad ($\tau = 10^{-5}$) | SGD ($m = 0.9$) |
| Client Optimizer[†] | SGD ($m = 0.9$) | SGD ($m = 0.9$) | SGD ($m = 0.9$) |
| Clients Per Round | 20 | 10 | 100 |
| Server Learning Rate | 0.5 | 5.0 | 0.5 |
| Client Learning Rate | 0.01 | 50.0 | 0.01 |
| Client Epochs | 10 | 5 | 20 |
| Burn In | 400 | 800 | 200 |
| | Client Inference Hyperparameters | | |
| Scale $\alpha_{\text{cov}}$[‡] | $5 \times 10^{-2}$ | $1 \times 10^{-8}$ | $5 \times 10^{-3}$ |
| MCMC Shrinkage | $1 \times 10^{-4}$ | $1 \times 10^{-6}$ | $1 \times 10^{-4}$ |
| Laplace Epochs | $5^\star$ | 5 | 5 |
| NGVI Epochs | 5 | 10 | 5 |
| NGVI Samples | 5 | 10 | 5 |
| NGVI $\beta_{\text{NGVI}}$ | 0.99 | 0.99 | 0.99 |
| | Client Inference Hyperparameters Search Space | | |
| Scale $\alpha_{\text{cov}}$ | $\{1, 2, 5, 10\} \times 10^{-2}$ | $1 \times \{10^{-7}, 10^{-8}, 10^{-9}\}$ | $\{1, 5\} \times \{10^{-2}, 10^{-3}, 10^{-4}\}$ |
| MCMC Shrinkage | | $1 \times \{10^{-3}, 10^{-4}, 10^{-5}, 10^{-6}\}$ | |
| Laplace Epochs | | $\{5, 10\}$ | |
| NGVI Epochs | | $\{5, 10\}$ | |
| NGVI Samples | | $\{5, 10\}$ | |
| NGVI $\beta_{\text{NGVI}}$ | | $\{0.9, 0.99\}$ | |

**Table 7:** Hyperparameters. [†]Client has two separate optimizers, one used in local optimization (SG-MCMC), and one used in local state updates (for stateful FedEP). When applied, the client state optimizer reuses the same configuration as the server optimizer. [‡]This is a per-data-point scale, and is also used in other approximate inference techniques. $^\star$The (stateful) FedEP uses 10 Laplace epochs.

| Method | Hyperparameter | | Accuracy (%, ↑) | | Rounds (#, ↓) | |
|---|---|---|---|---|---|---|
| | | | 1000R | 1500R | 45% | 50% |
| FedEP (I) | Scale $\alpha_{\text{cov}}$ | $4 \times 10^{-2}$ | 49.1 | 50.3 | 457 | 1081 |
| | | $5 \times 10^{-2}$ | 48.9 | 50.5 | 464 | 1105 |
| | | $6 \times 10^{-2}$ | 48.8 | 50.5 | 474 | 1206 |
| FedEP (M) | MCMC Shrinkage | $5 \times 10^{-5}$ | 47.8 | 48.8 | 482 | — |
| | | $1 \times 10^{-4}$ | 48.9 | 50.5 | 456 | 1179 |
| | | $5 \times 10^{-4}$ | 49.2 | 49.2 | 436 | — |
| FedEP (L) | Laplace Epochs | 5 | 46.7 | 47.9 | 513 | — |
| | | 10 | 46.7 | 47.9 | 514 | — |
| FedEP (V) | NGVI Epochs | 5 | 47.9 | 49.6 | 478 | — |
| | | 10 | 46.6 | 48.5 | 523 | — |
| FedSEP (I) | Scale $\alpha_{\text{cov}}$ | $4 \times 10^{-2}$ | 48.2 | 48.7 | 431 | — |
| | | $5 \times 10^{-2}$ | 48.3 | 49.0 | 431 | — |
| | | $6 \times 10^{-2}$ | 48.3 | 49.1 | 433 | — |
| FedSEP (M) | MCMC Shrinkage | $5 \times 10^{-5}$ | 48.3 | 48.9 | 431 | — |
| | | $1 \times 10^{-4}$ | 48.3 | 49.0 | 432 | — |
| | | $5 \times 10^{-4}$ | 48.4 | 49.0 | 431 | — |
| FedSEP (L) | Laplace Epochs | 5 | 47.2 | 47.9 | 437 | — |
| | | 10 | 47.2 | 47.9 | 439 | — |
| FedSEP (V) | NGVI Epochs | 5 | 47.9 | 48.8 | 432 | — |
| | | 10 | 47.8 | 48.6 | 432 | — |

**Table 8:** CIFAR-100 Hyperparameter Analysis Experiments. FedEP and FedSEP refer to the stateful EP and stateless stochastic EP. We use **I** (Scaled Identity Covariance), **M** (MCMC), **L** (Laplace), and **V** (NGVI) to refer to different inference techniques.

