# OpenReview forum: "Federated Learning as Variational Inference: A Scalable Expectation Propagation Approach"
_ICLR.cc/2023/Conference — ICLR 2023 poster_

### Official Review · Reviewer_Z7LG · 2022-10-20

**Confidence:** 3
**Correctness:** 3
**Technical Novelty And Significance:** 2
**Empirical Novelty And Significance:** 2
**Recommendation:** 6

**Clarity, Quality, Novelty And Reproducibility:**

Clarity: The paper is very well written, explaining EP in detail, and it is very didactic.
Quality: EP is a solid algorithm whose usefulness is proven on previous literature.
Novelty: Maybe another weak point, as here it is a little engineering enhancement of EP.
Reproducibility: Authors state that the paper will be shared upon acceptance.

**Strength And Weaknesses:**

Strength: Solid methodology adapting EP, clarity.
Weaknesses: Low empirical evidence.

**Summary Of The Paper:**

The paper proposes a curious algorithm to adapt federated learning in the variational inference framework. In particular, it uses a scalable version of EP to do so. The contribution seems solid although, given its practical application, I miss a real case scenario, for example in hospitals, of the algorithm.

**Summary Of The Review:**

First thanks for the paper, I have enjoyed its reading. I would like to emphasize here the clarity of this paper and its organization. It is very easy to read although the topic is complex. My main issue that justifies not giving it a better recommendation is that this approach seems to be a practical real-case scenario but the experiments and somehow limited. For instance, your example of hospitals is great, but then you do not apply it to a real case scenario (maybe stackoverflow is the more complicated one). It would be a great paper if a more complex experiment could be done.

Concerning technical details I find some decisions a little bit arbitrary. Why have you used EP and not some version of VI? (Justify this ;-)) Why do not you use EP in the client inference? You can also use it and it would be a full EP approach? Have you considered using PowerEP instead of EP? How can I determine or tune which is the optimal algorithm for the EP client inference? (It would be just great to use Bayesian optimization here to develop an auto-variational-inference approach here). And lastly you need to justify the dumping and whether the non-convergence of EP is a problem. It would also be great to talk about the complexity of your method and some drawbacks about this approach with respect to other variational inference approaches. This are basically all the questions that I have asked myself when reading this.

Some minor issues or enhancements to the paper are including a paragraph with the organization of the paper in the introduction, introduce a further work paragraph in the conclusions section as I believe that this opens new research lines, put the related work section (IMHO) before, better describe the results obtained by the toy experiment, include in section 1 a reference to other divergence functions instead of KL and in section 2 to variational families and justify why have you used an improper uniform prior (advantages and drawbacks of the decision).

---

> ### Author Response · Authors · 2022-11-14
> **Response**
>
> We thank the reviewer for the thoughtful review!
>
> ### Novelty
> We agree that our work is not necessarily novel from an "approximate inference methodology" standpoint. Indeed, EP has been around for a while, and has even been applied to FL in recent years as we note in the paper (Bui et al. 2018, Corinzia et al. 2019, Ashman et al. 2022). However, these papers applied EP to small-scale FL benchmarks. For example, the maximum number of clients considered by the above works is 100, and the largest model considered by the above works is a 2-layer MLP with 200 hidden units or a 1-layer LSTM with 100 units. In contrast, we consider a much larger scale setting, e.g., ResNet-18 models with 10M+ parameters on CIFAR, and hundreds of thousands of clients on StackOverflow. To our knowledge, our work is the first application of EP to modern federated learning benchmarks/models. We thus believe that our work is significantly novel from an empirical standpoint. It has moreover not been clear how to practically scale-up EP to larger neural networks and more clients. Our work has several methodological/experimental contributions to help practitioners in scaling up EP: interpretation of damping as adaptive updates, stochastic EP for stateless clients, experiments across different client inference methods, etc. More generally, we believe that describing strategies for scaling up existing and well-known methods is itself an important contribution.
>
> ### More complex experiments
>
> We agree it would be nice to apply this to an even more complex scenario! Indeed, this is one criticism of modern FL benchmarks which (still) do not fully reflect the complexity of distributed and private learning. However, we note that the benchmarks we consider (which were released by Google: https://www.tensorflow.org/federated) are still much more complex compared to previous works (as mentioned above), and are considered "standard" in modern FL.
>
> ### Why EP vs. other versions of VI
>
> We used EP instead of other VI approaches as EP naturally extends FedPA, another Bayesian approach to FL that has been found to perform well that serves as our main baseline. In particular, EP modifies the FedPA objective by taking into account the current global approximation during client inference (see section 2 "Relationship to Federated Learning via Posterior Averaging"). As Bayesian FL becomes more popular, it would be interesting to explore other versions of VI (or approximate inference methods).
>
> ### EP for client inference?
>
> Great question! Running full EP on the client side would require maintaining the model parameters for each data point (although other versions of EP can make things more efficient). We also wanted to make client inference as efficient as possible in order to make our approach comparable to standard non-Bayesian approaches (i.e., FedAvg). We thus resorted to gradient-based inference techniques on the client side.
>
> ### Power EP?
> This is an interesting suggestion. We did not consider PowerEP (which effectively uses the alpha divergence instead of KL) since this formulation makes gradient-based inference (which is important for maintaining efficiency) not straightforward. But it would be interesting to consider this in the future.
>
> ### Optimal algorithm for client inference
> We agree it would be amazing to use Bayesian optimization here :). Our experiments show that for realistic benchmarks, EP/SEP with scaled identity or SG-MCMC works well (section 3.3 discusses why this may be the case for scaled identity), despite being simple and efficient.
>
> ### Damping and Non-convergence
> Damping was particularly important to our setting since (1) clients send messages to the centralized server in parallel (instead of sequentially), and since we can also reinterpret damping as a gradient update to enable adaptive optimization methods (adaptive optimization is now considered standard for FL). We discuss the non-convergence of EP in section 2.1 and the "Limitations" part of section 3.3. But we agree that these are important points and we will add more discussion around these points in the next iteration of the paper.
>
> ### FedEP Complexity
> In the "Limitations" section we discuss the additional memory/compute requirements for our methods. In summary, Fed{S}EP with scaled identity and MCMC has the same compute requirements as regular FedAvg, while Fed{S}EP with Laplace/NVGI requires slightly more client compute to estimate the client covariances. Communication cost is doubled for MCMC/Laplace/NVGI due to the need to communicate the covariance vector, while the communication cost is the same between FedAvg and FedEP+scaled identity.
>
> ### Some minor issues...
> These are great suggestions! We will include the suggested changes/discussions in the paper.

---

> > ### Author Response · Authors · 2022-12-01
> > **Discussion**
> >
> > Hi there! Thanks again for your review. We believe we have addressed many of the points raised in the original review. We additionally wanted to emphasize our response to the "low empirical evidence" part of the review.
> >
> > In particular:
> > - The federated learning setup (models, datasets, etc.) that we consider is actually quite standard in the literature. For example, the [GitHub repo](https://github.com/google-research/federated/tree/master/posterior_averaging) on which our baselines are based has 450+ stars.
> > - In order to make sure our empirical results are robust, we conducted extensive experiments across different seeds and hyperparameters (which was also requested by Reviewer jmPj) to ensure that the outperformance of FedEP is not due to chance. The below table (and also the updated figures 2 and 3 in the paper) have the standard deviation metrics from running each method 5 times, while Table 8 in the appendix shows the robustness to the hyperparameters.
> >
> > **CIFAR-100**
> >
> > | Method   | Accuracy (1000R) | Accuracy (1500R) | Rounds (45\%) | Rounds (50\%) |
> > | ------- | ------- | ------- | ------- | ------- |
> > | FedPA                | $45.8$ | $48.4$ | $811$ | $-$ |
> > | FedAvg               | $44.7_{(0.2)}$ | $46.2_{(0.2)}$ | $911_{(86)}$ | $-$ |
> > | FedEP (I)            | $\underline{48.7}_{(0.4)}$ | $\textbf{50.7}_{(0.4)}$ | $\underline{473}_{(17)}$ | $\textbf{1167}_{(107)}$ |
> > | FedEP (M)            | $\textbf{48.8}_{(0.4)}$ | $\underline{50.4}_{(0.5)}$ | $\textbf{461}_{(13)}$ | $\underline{1240}_{(133)}^\dagger$ |
> > | FedEP (L)            | $46.5_{(0.4)}$ | $47.7_{(0.3)}$ | $523_{(28)}$ | $-$ |
> > | FedEP (V)            | $47.8_{(0.5)}$ | $49.6_{(0.6)}$ | $487_{(24)}$ | $1290_{(-)}^\ddagger$ |
> > | FedSEP (I)           | $48.2_{(0.4)}$ | $48.9_{(0.4)}$ | $438_{(9)}$  | $-$ |
> > | FedSEP (M)           | $48.2_{(0.4)}$ | $48.9_{(0.4)}$ | $438_{(9)}$  | $-$ |
> > | FedSEP (L)           | $47.2_{(0.4)}$ | $47.8_{(0.4)}$ | $442_{(10)}$ | $-$ |
> > | FedSEP (V)           | $47.8_{(0.4)}$ | $48.5_{(0.5)}$ | $440_{(10)}$ | $-$ |
> >
> > **StackOverflow**
> >
> > | Method | prec. | recall | mi-F1 | ma-F1 |
> > | ------- | ------- | ------- | ------- | ------- |
> > | FedPA                | $\underline{74.66}$ | $19.94$ | $30.78$ | $11.63$ |
> > | FedAvg               | $\textbf{75.20}_{(0.18)}$ | $13.88_{(0.27)}$ | $23.32_{(0.41)}$ | $8.02_{(0.26)}$ |
> > | FedSEP (I)           | $71.32_{(0.20)}$ | $25.10_{(0.22)}$ | $37.04_{(0.25)}$ | $13.61_{(0.15)}$ |
> > | FedSEP (M)           | $58.31_{(18.04)}$ | $8.70_{(1.14)}$ | $14.29_{(2.20)}$ | $2.70_{(0.33)}$ |
> > | FedSEP (L)           | $70.98_{(0.18)}$ | $\underline{25.88}_{(0.30)}$ | $\underline{37.80}_{(0.29)}$ | $\underline{13.97}_{(0.19)}$ |
> > | FedSEP (V)           | $69.51_{(0.35)}$ | $\textbf{28.02}_{(0.20)}$ | $\textbf{39.78}_{(0.25)}$ | $\textbf{15.32}_{(0.08)}$ |
> >
> > Please let us know if you would like to see any additional experiments!

---

### Official Review · Reviewer_jmPj · 2022-10-21

**Confidence:** 5
**Correctness:** 3
**Technical Novelty And Significance:** 1
**Empirical Novelty And Significance:** 2
**Recommendation:** 6

**Clarity, Quality, Novelty And Reproducibility:**

Clarity is high, quality is high with the exception of the lack of multiple seeds and hyper-parameter explanation. Novelty is not very high. Reproducibility seems good given Table 6. I feel confident I could replicate these experiments.

I see the lack of multiple seeds as a reason for rejection for this paper. Should the authors include those results across >=3 seeds, and should they provide at discussion of the hyperparameters, I will raise my score. I will further raise my score if the authors show an ablation study across the different hyperparameters of Table 6 in order to understand their influence on the different algorithms.

**Strength And Weaknesses:**

The strengths of this paper lie in the experimental settings and exposition.
Cifar100 with ResNet18 does serve as an example for model-scale (as far as bayesian methods are concerned), although SEP with (I) or maybe also (V) could be applied to even larger models. It would be interesting to see these methods compared against adaptive FedAVG on e.g. a wide resnet 50.
Stackoverflow experiments serve as a good example for a large number of clients with a simple LR model.
The exposition is good, explaining EP and its variants in sufficient details.

The weakness of this paper lie in the level of contribution to ICLR. None of the proposed algorithms is significantly novel as far as I can tell. This paper might be better suited as a workshop contribution.

Another weakness lies in that for an empirical study especially, I'd have expected more than a single seed for the proposed experiments. While most results seem significantly far apart based on my experience with these methods and data-sets, I would like to see standard-error across several seeds to judge significance. Learning curves show the strengths of EP in principle compared to FedAvg but don't allow to compare different methods and prior work (FedPA).

For an empirical effort, I would expect to see the search-space for the chosen Hyperparameters in Table 6 and a discussion on how (and if) the optimal hyperparameters chosen here generalize across the different algorithms. E.g. does FedEP(L) require the same hyper-parameters as FedSEP(V). Were these parameters tuned on the non-bayesian baseline? I can imagine that the stochasticity inherent to the bayesian optimization procedures warrants different hyper-parameters.



Small things:
Typo section 3.1: "Fig. 1 illustrates a simple case where posterior averaging [performs] sub-optimally."
Figure 4: What is the difference between the two FedEP runs in middle and right sub figure?
Tables 1-4, please improve readability, especially by separating captions between Table 2 & 4. I might suggest reducing the font-size across the tables.


**Summary Of The Paper:**

This paper presents an empirical study of different Expectation Propagation algorithmic choices applied to several federated learning settings. Important aspects to an empirical studies are missing.

**Summary Of The Review:**

Interesting analysis showing the applicability of EP in FL settings. The R18 and SOF setting is convincing wrt. the aspect of scale. The empirical evaluation is lacking at the moment wrt. seeds and hyperparameter selection.

---

> ### Author Response · Authors · 2022-11-17
> **Response (1/2)**
>
> Thanks to the reviewer for the comments!
>
> ### Novelty
> We agree that our work is not necessarily novel from an "approximate inference methodology" standpoint. Indeed, EP has been around for a while, and has even been applied to FL in recent years as we note in the paper (Bui et al. 2018, Corinzia et al. 2019, Ashman et al. 2022). However, these papers applied EP to small-scale FL benchmarks. For example, the maximum number of clients considered by the above works is 100, and the largest model considered by the above works is a 2-layer MLP with 200 hidden units or a 1-layer LSTM with 100 units. In contrast, we consider a much larger scale setting, e.g., ResNet-18 models with 10M+ parameters on CIFAR, and hundreds of thousands of clients on StackOverflow. To our knowledge, our work is the first application of EP to modern federated learning benchmarks/models. We thus believe that our work is significantly novel from an empirical standpoint. It has moreover not been clear how to practically scale-up EP to larger neural networks and more clients. Our work has several methodological/experimental contributions to help practitioners in scaling up EP: interpretation of damping as adaptive updates, stochastic EP for stateless clients, experiments across different client inference methods, etc. More generally, we believe that describing strategies for scaling up existing and well-known methods is itself an important contribution.
>
> ### Experiments with more (5) Seeds
> We followed the reviewer's suggestion and repeated the experiments with 4 additional seeds (5 in total). We have updated Tables 2 and 3, as well as Figures 2 and 3 in the paper correspondingly. We also attached both the updated tables and figures below for convenience (with standard deviations in brackets). Results indicate that the outperformance of FedEP is statistically significant.
>
> - Updated Figure 2: [img](https://i.imgur.com/NYIL2SL.png)
> - Updated Figure 3: [img](https://i.imgur.com/Z15I4lj.png)
> - Updated Table 2:
>
> | Method   | Accuracy (1000R) | Accuracy (1500R) | Rounds (45\%) | Rounds (50\%) |
> | ------- | ------- | ------- | ------- | ------- |
> | FedPA                | $45.8$ | $48.4$ | $811$ | $-$ |
> | FedAvg               | $44.7_{(0.2)}$ | $46.2_{(0.2)}$ | $911_{(86)}$ | $-$ |
> | FedEP (I)            | $\underline{48.7}_{(0.4)}$ | $\textbf{50.7}_{(0.4)}$ | $\underline{473}_{(17)}$ | $\textbf{1167}_{(107)}$ |
> | FedEP (M)            | $\textbf{48.8}_{(0.4)}$ | $\underline{50.4}_{(0.5)}$ | $\textbf{461}_{(13)}$ | $\underline{1240}_{(133)}^\dagger$ |
> | FedEP (L)            | $46.5_{(0.4)}$ | $47.7_{(0.3)}$ | $523_{(28)}$ | $-$ |
> | FedEP (V)            | $47.8_{(0.5)}$ | $49.6_{(0.6)}$ | $487_{(24)}$ | $1290_{(-)}^\ddagger$ |
> | FedSEP (I)           | $48.2_{(0.4)}$ | $48.9_{(0.4)}$ | $438_{(9)}$  | $-$ |
> | FedSEP (M)           | $48.2_{(0.4)}$ | $48.9_{(0.4)}$ | $438_{(9)}$  | $-$ |
> | FedSEP (L)           | $47.2_{(0.4)}$ | $47.8_{(0.4)}$ | $442_{(10)}$ | $-$ |
> | FedSEP (V)           | $47.8_{(0.4)}$ | $48.5_{(0.5)}$ | $440_{(10)}$ | $-$ |
>
> - Updated Table 3
>
> | Method | prec. | recall | mi-F1 | ma-F1 |
> | ------- | ------- | ------- | ------- | ------- |
> | FedPA                | $\underline{74.66}$ | $19.94$ | $30.78$ | $11.63$ |
> | FedAvg               | $\textbf{75.20}_{(0.18)}$ | $13.88_{(0.27)}$ | $23.32_{(0.41)}$ | $8.02_{(0.26)}$ |
> | FedSEP (I)           | $71.32_{(0.20)}$ | $25.10_{(0.22)}$ | $37.04_{(0.25)}$ | $13.61_{(0.15)}$ |
> | FedSEP (M)           | $58.31_{(18.04)}$ | $8.70_{(1.14)}$ | $14.29_{(2.20)}$ | $2.70_{(0.33)}$ |
> | FedSEP (L)           | $70.98_{(0.18)}$ | $\underline{25.88}_{(0.30)}$ | $\underline{37.80}_{(0.29)}$ | $\underline{13.97}_{(0.19)}$ |
> | FedSEP (V)           | $69.51_{(0.35)}$ | $\textbf{28.02}_{(0.20)}$ | $\textbf{39.78}_{(0.25)}$ | $\textbf{15.32}_{(0.08)}$ |

---

> > ### Author Response · Authors · 2022-11-18
> > **Response (2/2)**
> >
> >
> > ### Hyperparameters Search Space and Ablation Analysis
> > **Task Hyperparameters.**
> > - We borrowed the task hyperparameters (e.g., client/server optimization learning rates, number of client epochs) from Al-Shedivat et al., (2021)'s FedPA paper. We did not tune any of these hyperparameters for FedEP. This potentially disadvantages FedEP compared to FedPA. While it may be possible to get even better performance by tuning these parameters, we wanted to make our approach as close as possible to the baseline FedPA approach.
> > - In our experiments, we reused these same hyperparameters for FedEP and FedSEP across different client inference techniques.
> >
> > **Client Inference Hyperparameters.**
> > - We have updated the paper (Table 7 in the appendix) to include the hyperparameters search space for the different client inference methods.
> > - As the reviewer suggested, we conducted additional ablation experiments to understand their influence on the different algorithms (Table 8). (We were only able to do this for CIFAR due to the rebuttal time window). We have attached the table below for convenience. We find that FedEP is robust to many hyperparameters, e.g., hyperparameters that work well for FedEP(L) also work well for FedSEP(V).
> >
> >
> > | Method  | Hyperparameter | | Accuracy (1000R) | Accuracy (1500R) | Rounds (45\%) | Rounds (50\%) |
> > | ----- | ----- | ----- | ----- | ----- | ----- | ----- |
> > | FedEP (I) | Scale $\alpha_{\text{cov}}$ | $4 \times 10^{-2}$  | $49.1$ | $50.3$ | $457$ | $1081$ |
> > | FedEP (I) | Scale $\alpha_{\text{cov}}$ | $5 \times 10^{-2}$  | $48.9$ | $50.5$ | $464$ | $1105$ |
> > | FedEP (I) | Scale $\alpha_{\text{cov}}$ | $6 \times 10^{-2}$  | $48.8$ | $50.5$ | $474$ | $1206$ |
> > | FedEP (M) | MCMC Shrinkage | $5 \times 10^{-5}$ | $47.8$ | $48.8$ | $482$ | $-$ |
> > | FedEP (M) | MCMC Shrinkage | $1 \times 10^{-4}$ | $48.9$ | $50.5$ | $456$ | $1179$ |
> > | FedEP (M) | MCMC Shrinkage | $5 \times 10^{-4}$ | $49.2$ | $49.2$ | $436$ | $-$ |
> > | FedEP (L) | Laplace Epochs | $5 $ | $46.7$ | $47.9$ | $513$ | $-$ |
> > | FedEP (L) | Laplace Epochs | $10$ | $46.7$ | $47.9$ | $514$ | $-$ |
> > | FedEP (V) | NGVI Epochs | $5 $ | $47.9$ | $49.6$ | $478$ | $-$ |
> > | FedEP (V) | NGVI Epochs | $10$ | $46.6$ | $48.5$ | $523$ | $-$ |
> > | FedSEP (I) | Scale $\alpha_{\text{cov}}$ | $4 \times 10^{-2}$ | $48.2$ | $48.7$ | $431$ | $-$ |
> > | FedSEP (I) | Scale $\alpha_{\text{cov}}$ | $5 \times 10^{-2}$ | $48.3$ | $49.0$ | $431$ | $-$ |
> > | FedSEP (I) | Scale $\alpha_{\text{cov}}$ | $6 \times 10^{-2}$ | $48.3$ | $49.1$ | $433$ | $-$ |
> > | FedSEP (M) | MCMC Shrinkage | $5 \times 10^{-5}$    | $48.3$ | $48.9$ | $431$ | $-$ |
> > | FedSEP (M) | MCMC Shrinkage | $1 \times 10^{-4}$    | $48.3$ | $49.0$ | $432$ | $-$ |
> > | FedSEP (M) | MCMC Shrinkage | $5 \times 10^{-4}$    | $48.4$ | $49.0$ | $431$ | $-$ |
> > | FedSEP (L) | Laplace Epochs | $5 $ | $47.2$ | $47.9$ | $437$ | $-$ |
> > | FedSEP (L) | Laplace Epochs | $10$ | $47.2$ | $47.9$ | $439$ | $-$ |
> > | FedSEP (V) | NGVI Epochs | $5 $ | $47.9$ | $48.8$ | $432$ | $-$ |
> > | FedSEP (V) | NGVI Epochs | $10$ | $47.8$ | $48.6$ | $432$ | $-$ |
> >
> > ### Small things
> > - **Figure 4: What is the difference between the two FedEP runs in middle and right sub figure?**: Ah thanks for catching this! We have corrected Figure 4.
> > - **Typos/readability**: We have fixed the typo. We will make sure to improve the readability of Tables 1-4 (for the rebuttal version of the paper, everything is squeezed since we added standard deviation numbers).

---

> ### Comment · Reviewer_jmPj · 2022-11-23
> **Thanks for the rebuttal**
>
> I thank the authors for their rebuttal and their improved empirical evaluation. The additional seeds help strengthen the paper. In line with the other reviewers I see novelty as an issue, but have raised my score to reflect the additional contribution to the empirical evaluation.

---

> > ### Author Response · Authors · 2022-11-24
> > **Thanks!**
> >
> > Thanks :)
> >
> > In addition to the seed experiments, the review stated that you will consider further raising the score "if the authors show an ablation study across the different hyperparameters of Table 6 in order to understand their influence on the different algorithms." I believe we have conducted the requested ablation study in Table 8 of the appendix (also shown in response 2/2), where we find that FedEP is quite robust to the various hyperparameter choices. Does our ablation study across the hyperparameter search space satisfy the above point, or would you like for us to run additional ablations?

---

### Official Review · Reviewer_Hh8V · 2022-10-24

**Confidence:** 4
**Correctness:** 3
**Technical Novelty And Significance:** 2
**Empirical Novelty And Significance:** 2
**Recommendation:** 6

**Clarity, Quality, Novelty And Reproducibility:**

The paper is mostly clear.  It is not original. The results in the paper are not reproducible. Explanations are missing and code wasn't provided.

**Strength And Weaknesses:**

Strength:
* A Bayesian approach that in some sense generalizes previous approaches.
* An important aspect of the work is the emphasis on scaling the model, which I deem as very important for Bayesian models.

Weaknesses/Questions:
* Novelty. While I appreciate the contribution in terms of scalability I am not sure that it is enough. I do not see how the approach is different than previous methods that also used EP for FL (and specifically, [1], [2]). Can the authors comment on that?
Furthermore, the methods to scale the model are also common solutions. So overall it seems like the paper combines existing elements from previous studies. I will state though, that in my opinion, the bigger issue is with the experimental section which I will present next.
* Experiments. I think that there are several issues with this part.
  * First, information is missing. How did you split the data among clients? How were the hyper-parameters optimized? Was there a validation set?
  * Second, in the paper you stated that you chose the best accuracy based on a running average of 100 rounds. What does that mean exactly? Why not take the metric results based on the last round or the best round based on a validation set? As far as I know, FL papers do not use this approach.
  * Third, the baselines in this paper are not enough. I think the method should be compared to additional Bayesian methods (for example, [3]), and to leading methods in FL (and not only FedAvg and FedPA).
  * In Fig. 2 and Fig. 3, where are the lines of FedEP until 400 and 800 steps respectively? currently, it looks a little bit odd. Also, how can you explain the jump at those points? Did you use a scheduler with a lr drop at those points?
  * Finally, one of the motivations for being Bayesian is the ability to quantify uncertainty accurately. How is FedEP compared to other methods in that aspect?
* Minor.
  * Tables 1-4 could be organized better. Also, their numbering should be according to the reference in the text.
  * The colors of the lines in the figures are too similar in my opinion.

[1] Bui, T. D., Nguyen, C. V., Swaroop, S., & Turner, R. E. (2018). Partitioned variational inference: A unified framework encompassing federated and continual learning. arXiv preprint arXiv:1811.11206.

[2]  Ashman, M., Bui, T. D., Nguyen, C. V., Markou, E., Weller, A., Swaroop, S., & Turner, R. E. (2022). Partitioned Variational Inference: A framework for probabilistic federated learning. arXiv preprint arXiv:2202.12275.

[3] Achituve, I., Shamsian, A., Navon, A., Chechik, G., & Fetaya, E. (2021). Personalized Federated Learning with Gaussian Processes. Advances in Neural Information Processing Systems, 34, 8392-8406.

**Summary Of The Paper:**

The paper presents FedEP, a federated learning approach based on expectation propagation in which a global inference task is constructed as local inference tasks. The authors also present several ways to scale the model for modern neural networks. The authors compared their method to baselines on CIFAR-100, StackOverflow, and EMNIST-62.

**Summary Of The Review:**

I think that a Bayesian approach to FL is a good direction; however, in terms of novelty, and more importantly, the experiments, this paper can greatly be improved. I am willing to reevaluate my review based on the authors' response.

---

> ### Author Response · Authors · 2022-11-17
> **Response (1/2)**
>
> Thanks for the review!
>
> ### Novelty
> We agree that our work is not necessarily novel from an "approximate inference methodology" standpoint. Indeed, EP has been around for a while, and has even been applied to FL in recent years as we note in the paper (Bui et al. 2018, Corinzia et al. 2019, Ashman et al. 2022). However, these papers applied EP to small-scale FL benchmarks. For example, the maximum number of clients considered by the above works is 100, and the largest model considered by the above works is a 2-layer MLP with 200 hidden units or a 1-layer LSTM with 100 units. In contrast, we consider a much larger scale setting, e.g., ResNet-18 models with 10M+ parameters on CIFAR, and hundreds of thousands of clients on StackOverflow. To our knowledge, our work is the first application of EP to modern federated learning benchmarks/models. We thus believe that our work is significantly novel from an empirical standpoint. It has moreover not been clear how to practically scale-up EP to larger neural networks and more clients. Our work has several methodological/experimental contributions to help practitioners in scaling up EP: interpretation of damping as adaptive updates, stochastic EP for stateless clients, experiments across different client inference methods, etc. More generally, we believe that describing strategies for scaling up existing and well-known methods is itself an important contribution.
>
> ### Experimental Setup
> Thanks for these questions! As noted in section 3.2 of the paper,  our experimental setup is almost identical to Al-Shedivat et al., (2021), which itself follows the standard FL benchmark setup in terms of models/datasets/client splits. More specifically, we follow their [open-sourced repository](https://github.com/google-research/federated/tree/master/posterior_averaging) and [TensorFlow Federated](https://www.tensorflow.org/federated) for partitioning the data among clients and data splitting. We include hyperparameter details in Tables 7 and 8 (chosen hyperparameters, search space, and analysis). We also follow these works and evaluate performance via running averages. This is a more robust metric as a single-round metric could be noisy.
>
> ### Fig 2,3
> Here we follow Al-Shedivat et al. (2021) again and use FedAvg for 400/800 rounds as burn-in before switching to FedEP. This is why the lines for FedAvg/FedEP are identical up to 400/800 rounds.
>
>
> ### Discussion of [3]
> We thank the reviewer for suggesting this work. We have updated the related works to include a discussion. We believe this paper presents an interesting future direction to extend the proposed methods to personalized FL, but would like to point out some key differences.
> - [3] primarily considers personalized FL, in which each client learns a separate model. This is different from the setting we consider in this work (i.e., learning a global model in a federated fashion).
> - This work mainly conducts experiments with small models such as LeNet. Our work instead considers larger models with more clients. It is not immediately clear how the proposed methods could be extended to these larger settings as the method scales linearly (cubically in the worst case) with the number of data points.
>
> [3] Achituve, I., Shamsian, A., Navon, A., Chechik, G., & Fetaya, E. (2021). Personalized Federated Learning with Gaussian Processes.

---

> > ### Author Response · Authors · 2022-11-18
> > **Response (2/2)**
> >
> > ### Uncertainty quantification
> > This is a very interesting suggestion! As suggested by the reviewer (and as is standard in uncertainty quantification literature) we conducted additional experiments to examine calibration from the different approaches. On CIFAR we evaluate each model's Expected Calibration Error (ECE) with 15 bins,
> > $$
> > \operatorname{ECE} = \sum_i^{N_{\text{bins}}} b_i\left|\operatorname{accuracy}_i - \operatorname{confidence}_i\right|,
> > $$
> > where $\operatorname{accuracy}_i$ is the top-$1$ prediction accuracy in $i$-th bin, $\operatorname{confidence}_i$ is the average confidence of predictions in $i$-th bin, and $b_i$ is the fraction of data points in $i$-th bin. Bins are constructed in a uniform way in the $[0,1]$ range.
> > We consider accuracy and calibration from the resulting approximate posterior in two ways: (1) point estimation, which just uses the final model (i.e., MAP estimate from the approximate posterior) to obtain the output probabilities for each data point, and (2) marginalized estimation, which samples 10 models from the approximate posterior and averages the output probabilities to obtain the final prediction probability. Results are in Table 5 in the updated paper, and are also shown below. Notice that FedEP/FedSEP improves both the accuracy (higher is better) as well as expected calibration error (lower is better). We think that this is a significantly novel and interesting finding and thank the reviewer again for the suggestion.
> >
> > | Method   | Accuracy (Point Estimation) | Accuracy (Marginalized) | ECE-15 &darr;  (Point Estimation) | ECE-15 &darr;  (Marginalized) |
> > | ----- | ----- | ----- | ----- | ----- |
> > | FedAvg               | $45.7$ | $-$ | $19.6$ | $-$ |
> > | FedEP (I)            | $50.3$ | $48.6$ | $5.0$ | $7.5$ |
> > | FedEP (M)            | $50.6$ | $49.7$ | $5.6$ | $4.5$ |
> > | FedEP (L)            | $47.9$ | $48.0$ | $8.8$ | $6.6$ |
> > | FedEP (V)            | $49.9$ | $49.4$ | $5.9$ | $1.5$ |
> > | FedSEP (I)           | $48.8$ | $48.0$ | $10.2$ | $3.1$ |
> > | FedSEP (M)           | $48.8$ | $48.1$ | $10.3$ | $3.3$ |
> > | FedSEP (L)           | $47.7$ | $47.9$ | $9.7$ | $7.1$ |
> > | FedSEP (V)           | $48.3$ | $48.7$ | $9.5$ | $3.5$ |
> >
> > ### Minor points
> >
> > We will make the suggested changes!

---

> > > ### Comment · Reviewer_Hh8V · 2022-11-26
> > > **Response to Authors**
> > >
> > > I thank the authors for the response and the additional experiments. I still believe that the experimental section is lacking. First, the comparison to baseline is problematic, and second, the experimental setup feels odd to me and is not consistent with common setups in this field as far as I know. Nevertheless, I agree with the authors that it is important to come up with scalable Bayesian methods, and I am happy to see that the model is able to generate reasonable calibration. By the way, Why didn't you include the ECE of FedPA? Please include that in the final version. Also, please include the full discussion and the results in your response. When factoring all of the above I decide to raise the score by one to 6.

---

> > > > ### Author Response · Authors · 2022-12-01
> > > > **Thanks for the Response, and ECE of FedPA**
> > > >
> > > > Thanks for the response!
> > > >
> > > > We re-iterate that our setup is quite standard within FL. For example, the [GitHub repo](https://github.com/google-research/federated/tree/master/posterior_averaging) on which our baselines are based has 450+ stars.
> > > >
> > > > We followed your suggestion and computed the ECE of FedPA (not included earlier because of time constraints and also because we use the original FedPA implementation which is from a different repo). Note that FedPA only has point estimation as the paper's formulation does not explicitly compute the server covariance and instead works with its implicit representation (more specifically, it works with an estimate of the precision matrix times the mean vector).
> > > >
> > > > The full results are now given below.
> > > >
> > > > | Method | Accuracy (point estimation) | Accuracy (marginalized) | ECE-15 &darr; (point estimation) | ECE-15 &darr; (marginalized) |
> > > > | ------- | ------- | ------- | ------- | ------- |
> > > > | FedPA                | $48.1$         | $-$ | $13.6$         | $-$ |
> > > > | FedAvg               | $46.6_{(0.7)}$ | $-$ | $19.5_{(0.4)}$ | $-$ |
> > > > | FedEP (I)            | $50.8_{(0.4)}$ | $49.6_{(0.6)}$ | $4.9 _{(0.3)}$ | $7.9_{(0.2)}$ |
> > > > | FedEP (M)            | $50.5_{(0.5)}$ | $50.2_{(0.4)}$ | $5.9 _{(0.5)}$ | $4.6_{(0.4)}$ |
> > > > | FedEP (L)            | $47.7_{(0.5)}$ | $47.8_{(0.5)}$ | $8.8 _{(0.4)}$ | $6.6_{(0.4)}$ |
> > > > | FedEP (V)            | $49.7_{(0.5)}$ | $49.5_{(0.3)}$ | $5.9 _{(0.4)}$ | $2.2_{(0.5)}$ |
> > > > | FedSEP (I)           | $49.0_{(0.4)}$ | $48.5_{(0.4)}$ | $10.0_{(0.4)}$ | $3.4_{(0.3)}$ |
> > > > | FedSEP (M)           | $48.9_{(0.4)}$ | $48.6_{(0.4)}$ | $10.1_{(0.4)}$ | $3.5_{(0.3)}$ |
> > > > | FedSEP (L)           | $47.7_{(0.5)}$ | $47.8_{(0.5)}$ | $9.6 _{(0.6)}$ | $7.2_{(0.6)}$ |
> > > > | FedSEP (V)           | $48.5_{(0.4)}$ | $48.7_{(0.4)}$ | $9.3 _{(0.4)}$ | $3.7_{(0.4)}$ |
> > > >
> > > > Finally, as suggested we will make sure to include these discussions in the final version of the paper.

---

### Official Review · Reviewer_za1u · 2022-10-27

**Confidence:** 3
**Correctness:** 4
**Technical Novelty And Significance:** 3
**Empirical Novelty And Significance:** 3
**Recommendation:** 6

**Clarity, Quality, Novelty And Reproducibility:**

Clarity / Reproducibility:
- From the paper alone, it seems not easy to reproduce the results. In particular, the NGVI update is only shown for the precision/covariance matrix but not for the mean. NGVI algorithms typically require a lot of tricks to make work well in practical deep learning. Just by reading the paper, it is not clear to confirm whether the unimpressive performance of NGVI is due to a poor implementation or due to NGVI itself.  Perhaps the pseudo-code for NGVI can be provided in appendix, similar to MCMC Algorithm 2.
- For readers without a background in EP, the equation (3) may be hard to understand. Perhaps a few sentences could be added explaining the intuition between the update, i.e. explain why the minimizer in Eq. (3) will be a good approximation of p_k.
- Using a uniform distribution as a prior seems problematic -- especially if the parameter space is unconstrained. There is no uniform distribution on the real numbers. What is an "improper" uniform distribution?

Minor comments, typos, etc.:
- Algorithm 3: Serve Inference -> Server Inference
- In the NGVI update, it should be beta_NGVI instead of beta.


**Strength And Weaknesses:**

Strengths:
- I found the toy experiment in Section 3.1 to be very illuminating, along with Table 4. FedEP greatly improves over FedAvg and FedPA.
- The experiments on real data are also convincing, and the comparison of the difference approximate inference schemes was interesting to see.
- The overall algorithm is agnostic to the approximate inference method, so advances in approximate inference may further improve FedEP.

Weaknesses:
- Some parts of the paper were not clear / easy to read (see next section).
- The algorithm seems not too novel, since EP and the used approximate inference schemes are well-known in literature.
- It was a bit disappointing to see that a very "simple" approximate inference method (scaled identity) works best. This somehow suggests that the good performance may not be due to the EP formalism, but some other effects.  The paper would be much stronger (and I would increase my rating in that case), if it could be shown that more accurate approximate inference would lead to more accurate results. For example the accuracy of the approximate inference methods could be scaled up on small problems by doing a gradually more expensive MCMC estimation of the posterior, or my increasing the number of samples in the NGVI algorithm.

**Summary Of The Paper:**

The paper proposes to use EP together with approximate inference methods for federated learning. The approach shows promising results on toy examples and real data.  The method works with any approximate inference methods, and different choices are evaluated and compared.

**Summary Of The Review:**

The proposed FedEP algorithm is shown to perform well in practice on toy examples and real data.  The comparison of the different approximate inference methods is interesting, and highlights how the method can use any approximate inference scheme.  I believe this work will be interesting to the bayesian deep learning community as well as people working in federated learning. I am therefore leaning towards acceptance of this work, despite the outlined weaknesses.

---

> ### Author Response · Authors · 2022-11-17
> **Response**
>
> Thanks for the comments! Please see our response below.
>
> ### Novelty
> We agree that our work is not necessarily novel from an "approximate inference methodology" standpoint. Indeed, EP has been around for a while, and has even been applied to FL in recent years as we note in the paper (Bui et al. 2018, Corinzia et al. 2019, Ashman et al. 2022). However, these papers applied EP to small-scale FL benchmarks. For example, the maximum number of clients considered by the above works is 100, and the largest model considered by the above works is a 2-layer MLP with 200 hidden units or a 1-layer LSTM with 100 units. In contrast, we consider a much larger scale setting, e.g., ResNet-18 models with 10M+ parameters on CIFAR, and hundreds of thousands of clients on StackOverflow. To our knowledge, our work is the first application of EP to modern federated learning benchmarks/models. We thus believe that our work is significantly novel from an empirical standpoint. It has moreover not been clear how to practically scale-up EP to larger neural networks and more clients. Our work has several methodological/experimental contributions to help practitioners in scaling up EP: interpretation of damping as adaptive updates, stochastic EP for stateless clients, experiments across different client inference methods, etc. More generally, we believe that describing strategies for scaling up existing and well-known methods is itself an important contribution.
>
> ### Performance of scaled identity vs. more expensive inference methods
> We were also disappointed to see that scaled identity worked best! Per your advice, we conducted experiments across more expensive inference schemes (NGVI with more steps) on a smaller tractable setting. In particular, in Sec. A.3 we extend the experiments from Sec 3.3 by looking at the performance as we increase the complexity (a proxy of quality) of approximate inference techniques. We vary the number of iterations in NGVI from 1 (cheap) to 10 (expensive) epochs. We can observe in Figure 6 (also attached below) that as we increase NGVI's computations, the performance improves.
> **Experimental Results: Please see [this figure](https://i.imgur.com/pdovuIe.png).**
>
> Moreover, from an **uncertainty quantification** perspective, when we computed calibration error metrics suggested by reviewer Hh8V, we were excited to see that more expensive inference methods resulted in better calibrated models. Please see our response to reviewer Hh8V below, and also Table 5 in Appendix A.4.
>
> ### NGVI Algorithm
> We agree that the NVGI algorithm should be explained better. As suggested by the reviewer, we have included the pseudo-code for our NGVI in Algorithm 5 (appendix).
>
> ### Better Motivations for Equation 3
> Thanks for the suggestion! We agree that equation 3 could be motivated better. We have added the following modification to the paper.
>
> Expectation propagation (EP) constructs a posterior approximation through iterating local computations that refine factors that approximate the posterior contribution from each client. In this spirit, we would ideally like to solve the following localized version of Eq. 2, where we replace one of the factors with its corresponding approximating factor,
> $$
>     q_k^{\text{new}}(\boldsymbol{\theta})
>     = \underset{ q \in \mathcal{Q}}{\arg\min} \text{ }
>       D\bigg(p_k(\boldsymbol{\theta}) \text{ } {\color{red} p_{-k}(\boldsymbol{\theta})} \text{ }\|\|\text{ } q(\boldsymbol{\theta}) \text{ } {\color{red} p_{-k}(\boldsymbol{\theta})}\bigg),
>     \text{where } p_{-k}(\boldsymbol{\theta}) \propto \frac{p_\text{global}(\boldsymbol{\theta})}{p_k(\boldsymbol{\theta})}.
> $$
> Unfortunately, the right-hand side of the divergence is the intractable posterior we would like to approximate in the first place. Instead, EP solves the following problem (Eq. 3),
> $$
>     q_k^{\text{new}}(\boldsymbol{\theta})
>     = \underset{ q \in \mathcal{Q}}{\arg\min} \text{ }
>       D\bigg(p_k(\boldsymbol{\theta}) \text{ } {\color{red} q_{-k}(\boldsymbol{\theta})} \text{ }\|\|\text{ } q(\boldsymbol{\theta}) \text{ } {\color{red} q_{-k}(\boldsymbol{\theta})}\bigg),
>     \text{where } q_{-k}(\boldsymbol{\theta}) \propto \frac{q_\text{global}(\boldsymbol{\theta})}{q_k(\boldsymbol{\theta})}.
> $$
>
> ### Improper Uniform Prior
> Indeed, a uniform distribution over the real numbers does not integrate to 1, which is why it's called an improper prior. Improper priors are commonly used in Bayesian modeling (e.g., see "Improper prior" section of https://en.wikipedia.org/wiki/Prior_probability#Improper_priors). Our use of the improper uniform distribution is motivated by both computational convenience and standard convention. With an improper uniform prior, we do not have to take into account the prior distribution when performing client inference. Moreover, improper priors are commonly used in the EP literature, e.g., see  Vehtari et al. 2020. However it would be extremely interesting to investigate other prior distributions for future work!

---

> > ### Author Response · Authors · 2022-12-01
> > **Discussion**
> >
> > Hi there! Please let us know if the above response answered some of your questions in the original review, and please let us know if you would like to see more experiments!
> >
> > In particular, the main new experiments that address this review are:
> > - Showing that in small settings where we have enough compute to perform more expensive posterior inference, better variance approximation actually helps (Figure 6 in the appendix).
> > - Even in the regular setting where we cannot afford to do more expensive client inference, models that approximate variance sometimes have better calibration error. (E.g., marginalized estimation with FedEP NGVI does much better than FedEP Scaled Identity from a calibration standpoint, as shown in Table 5 of the appendix).

---

### Author Response · Authors · 2022-11-18
**General response**

We thank all the reviewers for their thoughtful reviews and comments. We have conducted additional experiments to address the questions posed by the reviewers and have updated the paper (changes from the previous version are in blue). While we have responded individually to each review, we also summarize our high-level changes here:

- **Uncertainty quantification (reviewer Hhv8)**: We calculated the expected calibration error from the different methods and found that FedEP models not only obtain higher accuracy, but also are calibrated better from an uncertainty quantification standpoint (i.e., less calibration error). This is shown in Appendix A.4 and Table 5 in the appendix of the updated paper.
- **Experiments with more seeds (reviewer jmPj)**: We ran all the CIFAR/Stackoverflow experiments with 5 seeds and have updated Figures 2, 3 and Tables 2, 3 to show the performance averaged across 5 runs along with the standard deviations. We find that the outperformance of FedEP is statistically meaningful.
- **Experiments across hyperparameters (reviewer jmPj)**: We have updated Table 7 in Appendix A.5 to show the hyperparameter search space for client inference. We additionally conducted ablation studies across these hyperparameters for CIFAR and have shown the results in Table 8 of the appendix. We find that FedEP is quite robust to the hyperparameters.
- **Experiments with more expensive inference in tractable setting (reviewer za1u)**: In Figure 6 of Appendix A.3 we have conducted experiments on a smaller tractable setting of CIFAR where we vary the number of NGVI inference steps to see if there is a setting where more accurate approximate inference does better than scaled identity. We do indeed find that when we are allowed to use more compute to improve client inference, NGVI does better than scaled identity.
- **Additional exposition**: We have also included additional changes (shown in blue of the updated paper) to improve clarity (e.g., addition of pseudo-code for NGVI algorithm in the appendix).

**Response regarding novelty.** We also wanted to make a general response to novelty. Several reviewers have commented that since EP is a well-known technique, our work lacks novelty. We agree that our work is not necessarily novel from an "approximate inference methodology" standpoint. Indeed, EP has been around for a while, and has even been applied to FL in recent years as we note in the paper (Bui et al. 2018, Corinzia et al. 2019, Ashman et al. 2022). However, these papers applied EP to small-scale FL benchmarks. For example, the maximum number of clients considered by the above works is 100, and the largest model considered by the above works is a 2-layer MLP with 200 hidden units or a 1-layer LSTM with 100 units.

In contrast, we consider a much larger scale setting, e.g., ResNet-18 models with 10M+ parameters on CIFAR, and hundreds of thousands of clients on StackOverflow. To our knowledge, our work is the first application of EP to modern federated learning benchmarks/models. We thus believe that our work is significantly novel from an empirical standpoint. It has moreover not been clear how to practically scale-up EP to larger neural networks and more clients. Our work has several methodological/experimental contributions to help practitioners in scaling up EP: interpretation of damping as adaptive updates, stochastic EP for stateless clients, experiments across different client inference methods, etc. More generally, we believe that describing strategies for scaling up existing and well-known methods is itself an important contribution.

---

### Decision · Program_Chairs · 2023-01-20

**Decision:**

Accept: poster

**Justification For Why Not Higher Score:**

The paper is a decent contribution to the area of Bayesian federated learning. However, though the paper does meet the acceptance bar, in my as well as the reviewers' opinion, the problem setting/solution is not that novel. The scalability aspect that the paper addresses is of course interesting, which led to the acceptance vote, and Accept (poster) would be the appropriate rating for the paper.

**Justification For Why Not Lower Score:**

The reviewers were unanimously leaning towards weak acceptance.

**Metareview: Summary, Strengths And Weaknesses:**

This paper looks at the problem of federal learning (FL) in the setting where we wish to infer the global posterior distribution as opposed to a global point estimate (traditional FL). The paper presents a method based on doing expectation propagation (EP) on each client. Although EP has been used in some prior work on federated learning, this paper develops a method which is more scalable in terms of number of parameters and number of clients. The techniques used to accomplish this are interesting and novel.

The reviewers initially had mixed opinions about the paper and some of the reviewers had concerns regarding novelty (in light of the existing work on EP in FL settings) and experiments (e.g., missing results on uncertainty quantification). The authors responded to the concerns and also provided additional experimental results during the rebuttal period, which addressed many of the concerns from the reviewers.

Post-rebuttal, all reviewers lean towards acceptance. From their reviews, discussions, and my own reading of the paper, we believe the paper has sufficient merits to be accepted. The authors are advised to incorporate the comments/suggestions from the reviewer in the final version.

**Note From Pc:**

if the above contains the word "oral" or "spotlight" please see: "oral" presentation means -> notable-top-5% and "spotlight" means -> notable-top-25%. As stated in our emails, we are disassociating presentation type from AC recommendations